

# Audible axions with a booster:
# Stochastic gravitational waves from rotating ALPs

Eric Madge[1][*], Wolfram Ratzinger[2][†], Daniel Schmitt[3][‡] and Pedro Schwaller[2][°]

**1** Department of Particle Physics and Astrophysics, Weizmann Institute of Science,
Rehovot 7610001, Israel
**2** PRISMA[+] Cluster of Excellence and Mainz Institute for Theoretical Physics,
Johannes Gutenberg-Universität Mainz, 55099 Mainz, Germany
**3** Institute for Theoretical Physics, Goethe Universität Frankfurt, 60438 Frankfurt, Germany

[*] eric.madge-pimentel@weizmann.ac.il , [†] w.ratzinger@uni-mainz.de ,
[‡] dschmitt@itp.uni-frankfurt.de , [°] pedro.schwaller@uni-mainz.de

## Abstract

Gravitational waves provide a novel way to probe axions or axion-like particles coupled to a dark photon field, even in the absence of couplings to Standard Model particles. In the conventional misalignment mechanism, the generation of an observable stochastic gravitational wave background, however, requires large axion decay constants. We here investigate the gravitational wave signal generated within the kinetic misalignment scenario, where the axion is assumed to have a large initial velocity. Its kinetic energy then provides a sufficiently high energy budget to generate a detectable gravitational wave signal also at lower values of the decay constant. We obtain an analytic estimate as well as perform numerical simulations of the corresponding gravitational wave signal, and evaluate its detectability at current and future gravitational wave observatories. We further present the corresponding projected constraints on the parameter space of the model, along with the parameter regions in which the dark photon or axion constitute dark matter, or in which the baryon asymmetry of the Universe is generated via the axiogenesis mechanism. Finally, we compute the GW production from the fragmentation of rotating axions, which is however difficult to observe experimentally.



# 1 Introduction

Little is known about the early Universe after the hot Big Bang and before the emission of the cosmic microwave background. Yet it could hold the key to understanding some of the most urgent open questions in particle physics, namely the origin and nature of dark matter (DM), and the source of the baryon asymmetry. Gravitational waves (GWs) are potential messengers from these early times. Future observations of primordial GWs will probe and constrain the phenomena that might have occurred in the early Universe, and therefore advance our understanding of the fundamental laws of nature.

The early Universe is very homogeneous and isotropic. Therefore, only rather violent processes can induce anisotropies in the energy-momentum tensor which are large enough to produce GWs that are still observable today.[1] In a series of papers [2–4] (see also Ref. [5]), we have shown that the dynamics of axions coupled to abelian gauge fields (aka dark photons) are viable sources of primordial GWs. There, the evolution of the axion field induces a tachyonic instability in the dark photon equations of motion. The resulting exponential growth of a range of dark photon modes amplifies its quantum fluctuations into macroscopic anisotropies which efficiently source GWs.

While originally proposed as a solution to the strong CP problem [6–9], where the QCD axion arises as the pseudo Nambu-Goldstone boson from the spontaneous breaking of the $U(1)_{\text{PQ}}$ Peccei-Quinn (PQ) symmetry, axions and axion-like particles (ALPs) generically occur

---

[1]For a recent, comprehensive review of potential cosmological sources, see e.g. Ref. [1] and references therein.

in many models of physics beyond the Standard Model (SM). They can potentially constitute DM [10–12], serve as the inflaton [13–19], emerge from string theory constructions [20–23], or dynamically solve the hierarchy problem [24]. As ALPs are typically assumed to interact only very weakly with the SM particles, gravitational probes may prove the most promising observational prospects for ALPs. As a consequence, a plethora of mechanisms for the generation of GWs within ALP models has been proposed [25–41], in particular including the so-called audible axion scenarios described above.

These audible axion scenarios produce an observable GW signal only for very large axion decay constants, $f_\phi \gtrsim 10^{17}$ GeV. Future GW experiments like LISA [42], Einstein Telescope [43] and pulsar timing arrays (PTAs) will therefore be able to probe otherwise inaccessible regions of the axion parameter space at highly suppressed couplings [3]. Detailed lattice computations have shown, however, that for a large range of axion masses, axion DM is over-produced, limiting the viability of the minimal audible axion scenario [4, 44–46].

It is straightforward to understand why large values of $f_\phi$ are needed to produce observable GWs. The fraction of the total energy density carried by the axion when it starts to roll is proportional to $f_\phi^2/M_{\rm Pl}^2$, where $M_{\rm Pl}$ is the Planck mass. There are essentially two possibilities to produce an observable signal for smaller $f_\phi$. First, if the onset of the axion evolution is delayed, its energy fraction increases relative to the radiation bath. This is for example realised in the relaxion model [40], where observable GWs are produced through the audible axion mechanism for $f_\phi$ as low as $10^8$ GeV. The second option is to equip the axion with a large amount of kinetic energy initially. In that case, both the axion mass and decay constant become relatively unconstrained, which furthermore makes it easy to obtain the correct axion DM abundance.

In this work we will focus on this second option, and use a large initial axion kinetic energy to obtain observable GWs across a broad range of parameters. Our work is based on the kinetic misalignment scenario [47–49], but should easily also apply to other models such as trapped misalignment [50, 51]. Kinetically misaligned axions are attractive for model building and phenomenology [52, 53], since they can explain the baryon asymmetry [47, 54–57] and modify the spectrum of long lasting primordial GW sources [58–60]. Furthermore, in combination with dark photons, GWs are produced along with vector dark matter. This was already noted in Ref. [61], where a rough estimate of the corresponding GW spectrum was presented. Here, we provide a more elaborate assessment of the GW background from kinetically misaligned audible axions. We compute the GW spectrum numerically and identify the regions of parameter space that may be probed by future GW experiments. We furthermore evaluate the cosmological constraints on the model and identify the regions where either the axion or the dark photon are viable DM candidates. Our main results are the GW spectrum shown in Fig. 4, and Figs. 8 and 9 which highlight the large range of viable axion DM parameter space that can be probed using GWs.

While the audible axion mechanism for GW production, in both, the conventional and kinetic misalignment scenario, relies on the presence of dark photons, GWs may also be sourced from excitations in the axion field itself due to the phenomenon of axion fragmentation [62–64]. We show that, indeed, GWs are hence generally produced in kinetic misalignment scenarios also in the absence of dark photons. As the resulting signal is, however, in general too small to be observable in the future, we defer a discussion of this scenario to the appendix.

## 2 Model Description

The following section gives an overview of the investigated model. This includes a short summary of the audible axion mechanism and a motivation for an extension by a finite initial ALP

velocity. In addition, we give a concrete computation of the axion kinetic energy that is generated via a higher-dimensional operator that explicitly breaks the $U(1)_{PQ}$ symmetry in the UV.

## 2.1 Audible Axions

In the audible axion mechanism, an ALP $\phi$ is introduced which is coupled to a dark gauge boson $X_\mu$ of an unbroken $U(1)_X$ symmetry group. The action of the model reads

$$\mathcal{S} = \int d^4x \sqrt{-g}\left[\frac{1}{2}\partial_\mu\phi\,\partial^\mu\phi - V(\phi) - \frac{1}{4}X_{\mu\nu}X^{\mu\nu} - \frac{\alpha}{4f_\phi}\phi X_{\mu\nu}\tilde{X}^{\mu\nu}\right], \tag{1}$$

where $f_\phi$ denotes the axion decay constant, hence the scale of $U(1)_{PQ}$ symmetry breaking that leads to the generation of the Nambu Goldstone boson $\phi$. The ALP couples to the dark electromagnetic field strength tensor $X_{\mu\nu}$ and its dual $\tilde{X}_{\mu\nu}$, with $\alpha$ denoting the dimensionless coupling constant. We assume the pseudoscalar is equipped with a potential of the form

$$V(\phi) = m_\phi^2 f_\phi^2\left[1 - \cos\left(\frac{\phi}{f_\phi}\right)\right], \tag{2}$$

which gives rise to the ALP mass $m_\phi$. In our previous work, we considered a minimal setup in terms of initial conditions, where through the misalignment mechanism the field $\phi$ is initially homogeneous and displaced from the minimum $\phi_0 = \theta f_\phi$, with $\theta = \mathcal{O}(1)$ denoting the misalignment angle. Furthermore, we assumed that there is no initial dark photon abundance, such that the field is in the Bunch-Davies vacuum. The field can then be written in terms of its mode functions $v_\lambda(k,\tau)$ as

$$\hat{X}^i(\mathbf{x},\tau) = \sum_{\lambda=\pm}\int\frac{d^3k}{(2\pi)^3}v_\lambda(k,\tau)\epsilon_\lambda^i(\mathbf{k})\hat{a}_\lambda(\mathbf{k})\exp(i\mathbf{k}\cdot\mathbf{x}) + \text{h.c.}, \tag{3}$$

where the creation and annihilation operators satisfy $\left[\hat{a}_\lambda(\mathbf{k}),\hat{a}_{\lambda'}^\dagger(\mathbf{k}')\right] = (2\pi)^3\delta_{\lambda\lambda'}\delta(\mathbf{k}-\mathbf{k}')$. We work in the Coulomb gauge $\nabla\cdot\mathbf{X} = 0$ and the circular polarization vectors obey the relations $\mathbf{k}\cdot\epsilon^\pm = 0$, $\mathbf{k}\times\epsilon^\pm = \mp ik\epsilon^\pm$, $\epsilon^\pm\cdot\epsilon^\pm = 0$, and $\epsilon^\pm\cdot\epsilon^\mp = 1$. Furthermore, the Bunch-Davies initial condition corresponds to the mode functions following $v_\lambda(k,\tau) = \exp(-ik\tau)/\sqrt{2k}$ at early times. We use $\tau$ to denote conformal time.

With the assumption of an homogeneous axion field, the equation of motion reads

$$\phi'' + 2aH\phi' + a^2\frac{\partial V}{\partial\phi} = \frac{\alpha}{f_\phi}a^2\mathbf{E}\cdot\mathbf{B}, \tag{4}$$

with $\mathbf{E}$ and $\mathbf{B}$ representing the dark electromagnetic fields. Primes denote derivatives with respect to conformal time $\tau$. The Hubble parameter $H = a'/a^2$ acts as a friction term, fixing $\phi$ at its initial value until $H \sim m_\phi$. As the expansion rate of the Universe drops below the mass of the pseudoscalar at the temperature $T_{\text{osc}} \approx \sqrt{m_\phi M_{\text{Pl}}}$, the ALP becomes free to roll down its potential and oscillations around the minimum start.

The nonzero axion velocity $\phi'$ causes an instability in the dark photon field. This can be seen by considering the equations of motion of the Fourier modes $v_\lambda(k,\tau)$ of the dark photon field

$$v_\pm''(k,\tau) + \omega_\pm^2 v_\pm(k,\tau) = v_\pm''(k,\tau) + \left(k^2 \mp k\frac{\alpha}{f_\phi}\phi'\right)v_\pm(k,\tau) = 0. \tag{5}$$

It is clear to see that as $\phi'$ becomes non-zero, the CP-violating axion-photon operator generates a tachyonic instability in the photon equation of motion for one helicity: Depending on the sign

of the axion velocity, the corresponding helicity exhibits a negative value of $\omega^2$, which leads to an exponential growth of dark vector modes $v_\pm \propto \exp(|\omega_\pm|\tau)$ in the range $0 < k < \alpha|\phi'|/f_\phi$. As a consequence, vacuum fluctuations are amplified into classical modes.

As shown in Refs. [2–4], the exponential production of dark photon quanta generates a stochastic gravitational wave background (SGWB) sourced by anisotropic, time-varying stress in the dark electromagnetic field. The resulting GW spectrum is peaked around the momentum scale that experiences the fastest growth. What makes the model particularly unique is its chiral nature, as the GW spectrum is strongly dominated by the photon helicity that is amplified first.

The model parameters controlling the GW peak amplitude and frequency are given by the decay constant $f_\phi$ and the ALP mass $m_\phi$, respectively. This can be understood by the fact that $f_\phi$ determines the initial amount of potential energy $\Omega_{\phi,\mathrm{osc}} \approx \left(\theta f_\phi/M_{\mathrm{Pl}}\right)^2$ that may be converted into gravitational radiation, while $m_\phi$ sets the temperature where axion oscillations start. Hence, by varying $m_\phi$, a SGWB may be generated over a large range of frequencies. However, for the GW spectrum to be detectable by future GW observatories, $f_\phi \gtrsim 10^{17}\,\mathrm{GeV}$ is necessary, making it impossible to directly detect the axion in the majority of viable parameter space. Furthermore, the dimensionless coupling is required to take rather large values of $\alpha = \mathcal{O}(100)$ in order to have efficient photon production, requiring additional model building [44, 65].

These considerations lead us to consider a finite initial velocity, since in this case the SGWB may be sourced exclusively by the kinetic energy of the pseudoscalar. In this scenario, a large parameter space in terms of $f_\phi$ opens up, as the energy budget available to be emitted via GWs becomes independent of the axion potential. This in turn allows for couplings $\alpha \ll 1$.

There is however one subtlety concerning this scenario: If the axion's kinetic energy dominates over the potential, the axions energy redshifts as $\rho_\phi = \frac{1}{2a^2}\phi'^2 \propto a^{-6}$. The typical dark photon growth rate therefore scales as $\omega \propto \frac{\alpha}{f}\phi' \propto a^{-2}$. To see whether tachyonic production is efficient, this rate needs to be compared to the comoving Hubble rate, which redshifts during radiation domination as $aH \propto a^{-1}$. Since the Hubble rate decreases more slowly, tachyonic production is either efficient right away or never, allowing for no sensible $\tau \to 0$ limit. This goes to show that the dynamics of dark photon production cannot be studied independently of the process causing the initial velocity. We therefore study a concrete implementation in the following.

## 2.2 Generation of finite ALP Velocity

Kinetic misalignment was proposed in [47, 48] as a mechanism to generate a finite ALP velocity. This scenario is inspired by Affleck-Dine baryogenesis [66, 67], where rotations of scalar particles are induced via higher-dimensional operators. To begin with, we identify the axion $\phi = \theta S$ as the angular component of a complex scalar field

$$P = \frac{1}{\sqrt{2}}\, S \exp(i\theta). \tag{6}$$

The radial component $S$ is called saxion[2] in the following and determines the effective decay constant $f_{\mathrm{eff}} = S$, which is identical to the ALP decay constant $f_\phi$ when $S$ takes its vacuum expectation value (VEV) at $\langle S \rangle = f_\phi$. As a concrete realization, we choose a quartic potential

$$V(P) = \lambda^2 \left(|P|^2 - \frac{f_\phi^2}{2}\right)^2 + V_{\cancel{PQ}}, \tag{7}$$

---

[2]Despite following the common nomenclature, we do not assume supersymmetry in this work.

for the field $P$, where the coupling constant is defined as $\lambda = m_{S,0}/\left(\sqrt{2}f_\phi\right)$, with $m_{S,0}$ being the vacuum mass of the saxion. We assume that the $U(1)_{\mathrm{PQ}}$ symmetry is explicitly broken in the UV by the higher-dimensional operator

$$V_{\not{\mathrm{PQ}}} = \frac{AP^n}{nM_{\mathrm{Pl}}^{n-3}} + \mathrm{h.c.} \, . \tag{8}$$

Here, $A$ denotes the dimensionful coupling and $n$ gives the mass dimension. These terms may be motivated by quantum gravitational effects at high energies for instance, or if the $U(1)_{\mathrm{PQ}}$ symmetry is generated as an accidental symmetry via other exact symmetries. The crucial point is that $V_{\not{\mathrm{PQ}}}$ generates an angular gradient in the potential at large field values, which may induce a rotation of $P$ that is related to a PQ charge density

$$n_{\mathrm{PQ}} = i\dot{P}^*P - i\dot{P}P^* = S^2\dot{\theta} \, . \tag{9}$$

Hence, the angular motion corresponds to a non-zero ALP kinetic energy. As the impact of the higher-dimensional term vanishes rapidly due to cosmic expansion, the $U(1)_{\mathrm{PQ}}$ symmetry is effectively restored as the Universe cools down. Due to charge conservation, the ALP continues to rotate around its potential. The energy density stored in the rotation then gives the available energy budget to be transferred to the dark gauge boson and eventually converted into gravitational radiation.

## 2.3 Initial Conditions

In this section, we provide the initial conditions of our model, including a calculation of the axion kinetic energy. First, we assume that the radial component $S$ is driven to large field values during inflation. This is a valid assumption if the quartic potential is sufficiently flat or $m_{S,0} \ll H_I$, with $H_I \leq 6 \times 10^{13}\,\mathrm{GeV}$ being the maximum Hubble scale during inflation [68]. Since this is given in the entire parameter space we consider, we take the saxion to be initially displaced from its minimum at a value $S_i$, which we treat as a free parameter in the following. In a radiation dominated universe, $P$ starts to roll when the Hubble parameter decreases to the order of the effective saxion mass at $S_i$, which reads

$$m_{S_i} = \sqrt{\left.\frac{\partial^2 V}{\partial S^2}\right|_{S=S_i}} = \sqrt{3}\lambda S_i \, . \tag{10}$$

Assuming that the saxion becomes free to oscillate when $3H = m_{S_i}$, we find

$$T_{S_i} = \left(\frac{30}{g_{\epsilon,S_i}\pi^2}\right)^{1/4}\sqrt{\lambda S_i M_{\mathrm{Pl}}} \tag{11}$$

as the temperature that marks the start of the evolution.

The key quantity to compute is the angular velocity that arises via the angular gradient induced by $V_{\not{\mathrm{PQ}}}$. In order to do so, we follow the approach from Refs. [47,48] and introduce the quantity

$$\epsilon \equiv \frac{n_{\mathrm{PQ}}}{n_S} \, , \tag{12}$$

that parameterises the ratio of the charge density in the rotation and the saxion number density. Hence, $\epsilon$ gives a measure of the shape of the path that $P$ follows, with $\epsilon = 1$ corresponding to a purely circular trajectory. The initial saxion number density is given by

$$n_{S_i} = \frac{V_0(S_i)}{m_{S_i}} \, , \tag{13}$$

where $V_0(S_i)$ denotes the initial potential energy. Thus, together with Eq. (9) the axion velocity right after the kick by $V_{\not{PQ}}$ may be expressed as

$$\dot{\phi}_{S_i} = \frac{\epsilon}{4\sqrt{3}}\lambda S_i^2.\tag{14}$$

It can be shown that $\epsilon$ is related to the dimensionful coupling $A$ and the mass dimension $n$ of $V_{\not{PQ}}$. However, in this work we set $\epsilon = 1$ for simplicity, hence we assume that $P$ exhibits a circular motion right after $T_{S_i}$, with the angular velocity being conserved up to cosmic expansion, as the impact of $V_{\not{PQ}}$ vanishes rapidly.

## 2.4 Dynamics

Before investigating the process of dark photon and gravitational wave production, it is worth to study the scaling behaviour of the system during the different stages of the evolution. As long as $S \gg f_\phi$, $P$ rotates in a quartic potential, hence mimicking the scaling behaviour of radiation, and it follows that

$$\dot{\phi} \propto a^{-2}, \quad \rho_\phi \propto a^{-4}, \quad S \propto a^{-1}.\tag{15}$$

During this phase of the evolution, $\Omega_\phi = $ const. up to changes in the relativistic degrees of freedom, unless photon production becomes effective. As the value of the saxion field decreases while the Universe expands, the radius of the circular trajectory approaches the ALP decay constant. When $S = f_\phi$, $P$ enters the minimum of the potential. We then obtain a kination-like scaling behaviour

$$\dot{\phi} \propto a^{-3}, \quad \rho_\phi \propto a^{-6}, \quad S = \text{const.}\tag{16}$$

From this moment, the radial degree of freedom takes a constant value. We thus regard the ALP independently. The cosine-like ALP potential as given by Eq. (2) then corresponds to a tilt of the total potential from Eq. (7). When the kinetic energy of the ALP becomes comparable to the height of the potential barriers $V_{\text{max}} = 2m_\phi^2 f_\phi^2$, the system enters a phase of matter-like scaling. Hence, the respective scale factor dependencies read

$$\dot{\phi} \propto a^{-3/2}, \quad \rho_\phi \propto a^{-3},\tag{17}$$

when the axion is trapped and behaves like DM as oscillations around the minimum start.

## 2.5 Model Constraints

As the mechanism is supposed to be embedded in the radiation dominated era of the Universe, we derive the condition for the radial component to make up a subdominant part of the total energy density. Right before $P$ starts to roll, the energy density stored in the radial component is given by $\rho_S \approx \lambda^2 S_i^4/4$. This should be smaller than the energy density of the SM plasma

$$\rho_{\text{tot}} = 3H^2 M_{\text{Pl}}^2 = \lambda^2 S_i^2 M_{\text{Pl}}^2,\tag{18}$$

where in the second step we use Eq. (11). It follows that

$$S_i < 2M_{\text{Pl}}.\tag{19}$$

This bound is met throughout the entire parameter space we consider in this work, hence it is a valid assumption to neglect the impact of the involved energy densities on the background evolution. Additionally, we may define the parameter range where kinetic misalignment is

active. That is the case if the kinetic energy of the rotation dominates over the height of the potential barriers when the axion enters the bottom of the Mexican hat:

$$\frac{1}{2}\dot{\phi}_{S_i}^2\left(\frac{a_{S_i}}{a_{S=f_\phi}}\right)^4 > 2m_\phi^2 f_\phi^2 \,, \tag{20}$$

where $a_{S=f_\phi}$ denotes the redshift when the saxion has rolled down to its VEV

$$\frac{a_{S_i}}{a_{S=f_\phi}} = \frac{f_\phi}{S_i} \,. \tag{21}$$

With the use of Eq. (14), this constraint may be translated to $\epsilon m_{S,0} \gtrsim 20 m_\phi$ under the assumption that the tachyonic window has not opened yet, since dark photon production would act as friction decreasing the ALP velocity.

Before including the dark vector dynamics in the next section, let us comment on the validity of the axion-dark photon coupling within the present framework. An effective operator as given in Eq. (4) may be generated by integrating out a heavy fermion $\psi$ charged both under $U(1)_X$ and $U(1)_{\text{PQ}}$,

$$\mathcal{L} \supset y_\psi P \bar{\psi} \psi + \text{h.c.} \,. \tag{22}$$

By requiring that thermal fermion production is inaccessible by the time of saxion oscillations, we find

$$y_\psi > \left(\frac{30}{g_{\epsilon,S_i}\pi^2}\right)^{1/4}\sqrt{\frac{\lambda M_{\text{Pl}}}{S_i}} \,. \tag{23}$$

On the other hand, the Hubble rate at the end of inflation $H_I$ is constrained by CMB measurements [68], which relates to the model parameters as

$$H_{S_i} = \frac{\lambda S_i}{\sqrt{3}} \leq H_I \,. \tag{24}$$

To make sure that possible quantum corrections on $\lambda$ do not violate this bound, we demand [61]

$$y_\psi^4 \lesssim 16\pi^2\lambda^2 \,. \tag{25}$$

Combining these constraints we obtain

$$\left(\frac{30}{16\pi^4 g_{\epsilon,S_i}}\right)^{1/4}\left(\frac{M_{\text{Pl}}}{S_i}\right)^{1/2} < 1 \,, \tag{26}$$

which gives a lower bound on the initial saxion field value, with a weak dependence on the saxion mass through the energetic degrees of freedom at the time the saxion starts to roll. Above the corresponding value of $S_i$, we can find a Yukawa coupling for which the fermion masses are large enough to avoid thermal production while at the same time small enough to neglect their quantum corrections to the saxion potential.

## 3 Dark Photon Production

In this section, we introduce the dark photons under the assumption of a finite ALP velocity. In particular, we study the conditions for successful tachyonic growth and give an analytic estimate of their growth time. In addition, we provide the results of our numerical simulation.

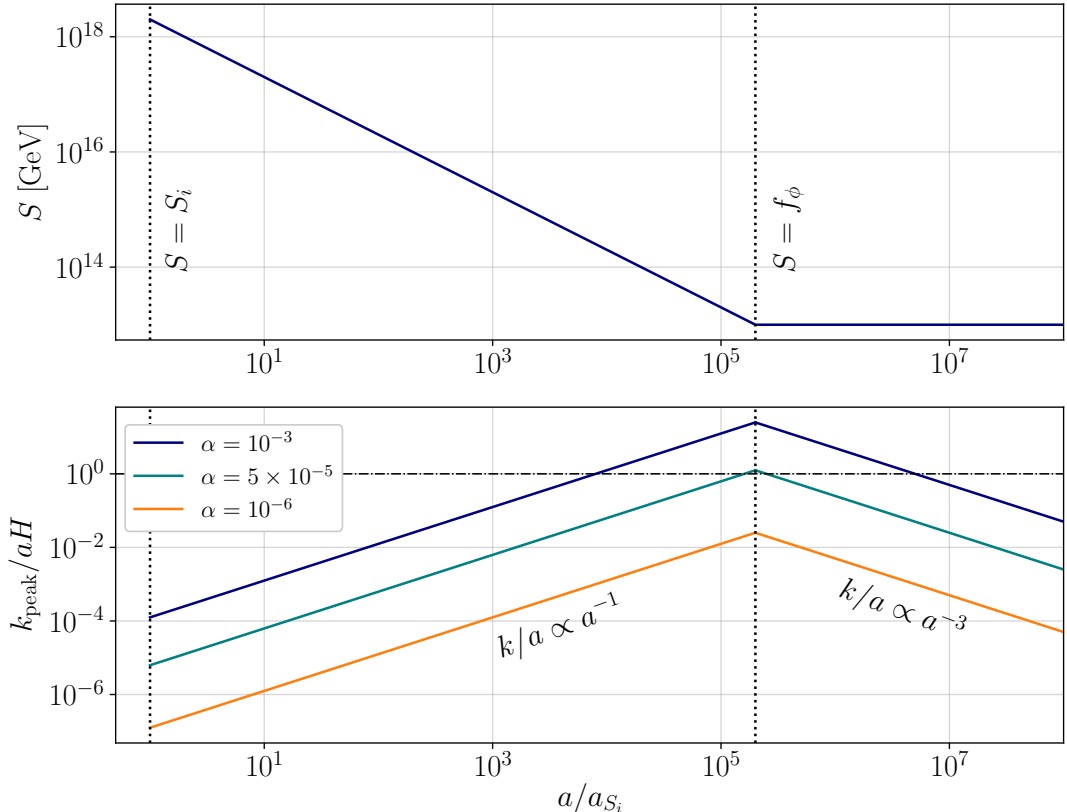

Figure 1: In the upper panel the evolution of the radial component is shown, starting out at $S_i = 2 \times 10^{18}$ GeV (left black dotted line) and finally settling at $S = f_\phi = 10^{13}$ GeV once the scale factor has grown to $a_{S=f_\phi}$ (right black dotted line). The lower plot shows the ratio of the dark photon growth rate $k_{\text{peak}}/a$ and the Hubble rate during radiation domination. Once this ratio surpasses unity (black dash-dotted line), tachyonic production of photons becomes efficient. This is only the case for large enough $\alpha$, with the green line being close to the limiting case, where $\alpha \approx \alpha_{\min}$.

## 3.1 Tachyonic Instability

Now that we know the dynamics of the saxion and axion respectively, we can estimate the conditions for efficient dark photon production. To do so, we generalise Eq. (5) by taking into account that the radial component of the complex field $P$ is no longer fixed at $f_\phi$

$$v''_\pm(k,\tau) + \left(k^2 \mp k\frac{\alpha}{S}\phi'\right)v_\pm(k,\tau) = 0. \tag{27}$$

Just like in the standard audible axion case discussed above, the fastest growing mode and the corresponding comoving growth rate are given by $k_{\text{peak}}(\tau) = \frac{\alpha}{2S(\tau)}|\phi'(\tau)|$. For tachyonic production to be efficient, the physical rate $k_{\text{peak}}/a$ needs to be larger than the Hubble rate $H$. While the saxion rolls down the quartic part of the potential, we have $S \propto a^{-1}$ and $\dot{\phi} \propto a^{-2}$ such that the growth rate scales as

$$k_{\text{peak}}/a = \frac{\alpha}{2S(\tau)}|\dot{\phi}(\tau)| = \frac{\alpha}{2S_i}|\dot{\phi}_{S_i}|\frac{a_{S_i}}{a} \propto a^{-1}, \tag{28}$$

which is slower than the Hubble rate during radiation domination, $H \propto a^{-2}$. On the other hand, once the saxion has settled in its minimum with $S = f_\phi$, the kination scaling sets in and

the growth rate is diminishing faster than the Hubble rate

$$k_{\text{peak}}/a = \frac{\alpha}{2f_\phi}|\dot{\phi}(\tau)| = \frac{\alpha}{2f_\phi}|\dot{\phi}_{S=f_\phi}|\left(\frac{a_{S=f_\phi}}{a}\right)^3 \propto a^{-3}. \tag{29}$$

Using the scaling of the Hubble rate $H$, that additionally takes into account changes in degrees of freedom, we can calculate the scale factor $a_*$ at which the tachyonic window opens up

$$\frac{a_*}{a_{S_i}} = \frac{8}{\alpha\epsilon}\left(\frac{g_{\epsilon,*}}{g_{\epsilon,S_i}}\right)^{1/2}\left(\frac{g_{s,S_i}}{g_{s,*}}\right)^{2/3}, \tag{30}$$

where $g_\epsilon$ and $g_s$ denote the effective degrees of freedom with respect to energy and entropy at the corresponding times. Since $\epsilon \leq 1$ and the fine structure constant is also expected to be small, we find that, initially, there is no tachyonic photon production. We can then distinguish three cases:

1. $a_* < a_{S=f_\phi}$: Dark photon production becomes efficient before the saxion takes on its VEV. In this case we expect efficient production at least until $S \approx f_\phi$, when the growth rate starts diluting faster than the Hubble rate.

2. $a_* \sim a_{S=f_\phi}$: In this case we only expect a very short period of tachyonic particle production, since, right after the window opens, it closes again due to the onset of the kination regime. Whether an $\mathcal{O}(1)$ fraction of the axion energy can be transmitted to the dark photon, which is required for GW emission, strongly depends on the exact time it takes the photon modes to grow, which we will study in the next section.

3. $a_* > a_{S=f_\phi}$: In this case the phase in which the growth rate increases relative to the Hubble rate is too short, such that kination sets in before the tachyonic window opens. Therefore, the production of photons is never efficient and only axion fragmentation might take place, which we will comment on in Appendix A.

We illustrate these cases in Fig. 1, where we leave all model parameters fixed and only vary $\alpha$. We observe that for $\alpha$ bigger than a threshold $\alpha_{\min}$ the tachyonic window opens. This threshold can be found from $a_* = a_{S=f_\phi}$ to be

$$\alpha_{\min} = \frac{8}{\epsilon}\left(\frac{g_{\epsilon,*}}{g_{\epsilon,S_i}}\right)^{1/2}\left(\frac{g_{s,S_i}}{g_{s,*}}\right)^{2/3}\frac{f_\phi}{S_i}. \tag{31}$$

Since $f_\phi \ll S_i$ and with $\epsilon = \mathcal{O}(0.1-1)$ there is a large parameter space in the kinetic misalignment scenario, where tachyonic production is possible without requiring $\alpha > 1$ as in the original audible axion scenario. Although large values of $\alpha$ can be achieved as shown in Refs. [44, 65], the small value of $\alpha$ allows for simpler UV completions.

## 3.2 Growth Time

So far we have only discussed when tachyonic production of dark photons starts. For efficient GW production it is however necessary that the majority of the axion energy is transferred to the photon. Since we assume there is initially no photon population except for vacuum fluctuations, there elapses some time between the onset of tachyonic production at $\tau_*$ and the emission of GWs at $\tau_{\text{GW}}$.

While the dark photon is in the Bunch-Davies vacuum, its mode functions are simply given as $v_\lambda(k, \tau < \tau_*) = 1/\sqrt{2k}\exp(-ik\tau)$, and therefore the energy in the resonance band is

$$\rho_{X,*} = \frac{1}{a_*^4}\frac{1}{4\pi^2}\int\limits_0^{2k_{\text{peak}}} dk\, k^2\left(|v_\lambda'(k, \tau_*)|^2 + k^2|v_\lambda(k, \tau_*)|^2\right) = \frac{1}{a_*^4}\frac{k_{\text{peak},*}^4}{16\pi^2}. \tag{32}$$

This energy needs to grow up to $\rho_{\phi,*} = \frac{1}{2}\dot{\phi}_*^2$ before GWs are emitted. Since the energy is proportional to the square of the mode functions, its growth rate is twice as big and we find the elapsed conformal time before most GWs are emitted,

$$\frac{\delta\tau}{\tau_*} = \frac{a_*H_*}{2k_{\text{peak},*}}\log\left(\frac{\rho_{\phi,*}}{\rho_{X,*}}\right) = \frac{1}{2}\log\left(\frac{\rho_{\phi,*}}{\rho_{X,*}}\right). \tag{33}$$

If the Universe is dominated by a radiation bath with changing degrees of freedom, this can be re-expressed as the redshift

$$\frac{a_{\text{GW}}}{a_*} = \left(\frac{g_{s,*}}{g_{s,\text{GW}}}\right)^{2/3}\left(\frac{g_{\epsilon,\text{GW}}}{g_{\epsilon,*}}\right)^{1/2}\left(1 + \frac{\delta\tau}{\tau_*}\right). \tag{34}$$

In Fig. 2 we show the results of a simulation that was started at $a_*$, right as the dark photon production becomes efficient. We can see the growth rate $k_{\text{peak}}/a$ starting to dominate over the Hubble rate. Around $a_{\text{GW}}$, as calculated with the formula above and marked by the black dotted line, the growth rate deviates from the analytic estimate, as a result of the axion slowing down due to the friction from dark photon production. This effect also becomes apparent in the bottom panel, where we can see the dark photons energy dominating over the axion soon after $a_{\text{GW}}$. After this point, the growth rate oscillates. Its mean, however, takes on a constant ratio with respect to the Hubble rate. This can be understood as the growth rate regulating itself: A large growth rate results in more efficient dark photon production, more friction on the axion and therefore a decrease in the growth rate.

Eventually, the saxion settles down at its VEV $f_\phi$ at $a_{S=f\phi}$ marked by the black, dash-dotted line. From here on out, the dark photon growth rate starts to decrease compared to the Hubble, due to the stronger effect of Hubble friction during kination scaling, and dark photon production becomes ineffective. As a consequence, we can see the energy densities taking on the expected scaling behaviors before we eventually stop the simulation, once the growth rate becomes smaller than the Hubble.

In Fig. 3 we show the evolution of the corresponding dark photon spectrum. At early times the modes near $k_{\text{peak}} = a_*H_* = a_{\text{GW}}/a_* \times a_{\text{GW}}H_{\text{GW}}$ grow the fastest and set the peak of the final spectrum. As the axion slows down due to the friction from photon production, only modes with smaller momenta are produced efficiently. Eventually, the saxion settles at its VEV $f_\phi$ and photon production becomes inefficient, leading to a suppression of the spectrum at low momenta.

Before we conclude this section, let us briefly comment on the possible cosmological observation of the dark photon, a free-streaming and non-interacting form of radiation. Limits on such forms of radiation are given as deviations of the effective number of neutrino-species $\Delta N_{\text{eff}} \propto \rho_X$. If we make the simplifying assumption that all of the axion's energy is transferred to the dark photon, we find

$$\Delta N_{\text{eff}} \simeq 9 \times 10^{-2} \times \frac{g_{\epsilon,S_i}}{g_{\epsilon,\text{rec}}}\left(\frac{g_{s,\text{rec}}}{g_{s,S_i}}\right)^{4/3}\left(\frac{\epsilon S_i}{M_{\text{Pl}}}\right)^2. \tag{35}$$

Therefore, as long as the constraints from the previous section are fulfilled and $\epsilon S_i$ does not exceed $M_{\text{Pl}}$, all current constraints on $\Delta N_{\text{eff}} \lesssim 0.3$ remain untouched [69].

In principle also the GWs that are discussed in detail below contribute to $\Delta N_{\text{eff}}$. Since their energy is always suppressed compared to the dark photons, this contribution can be neglected though.

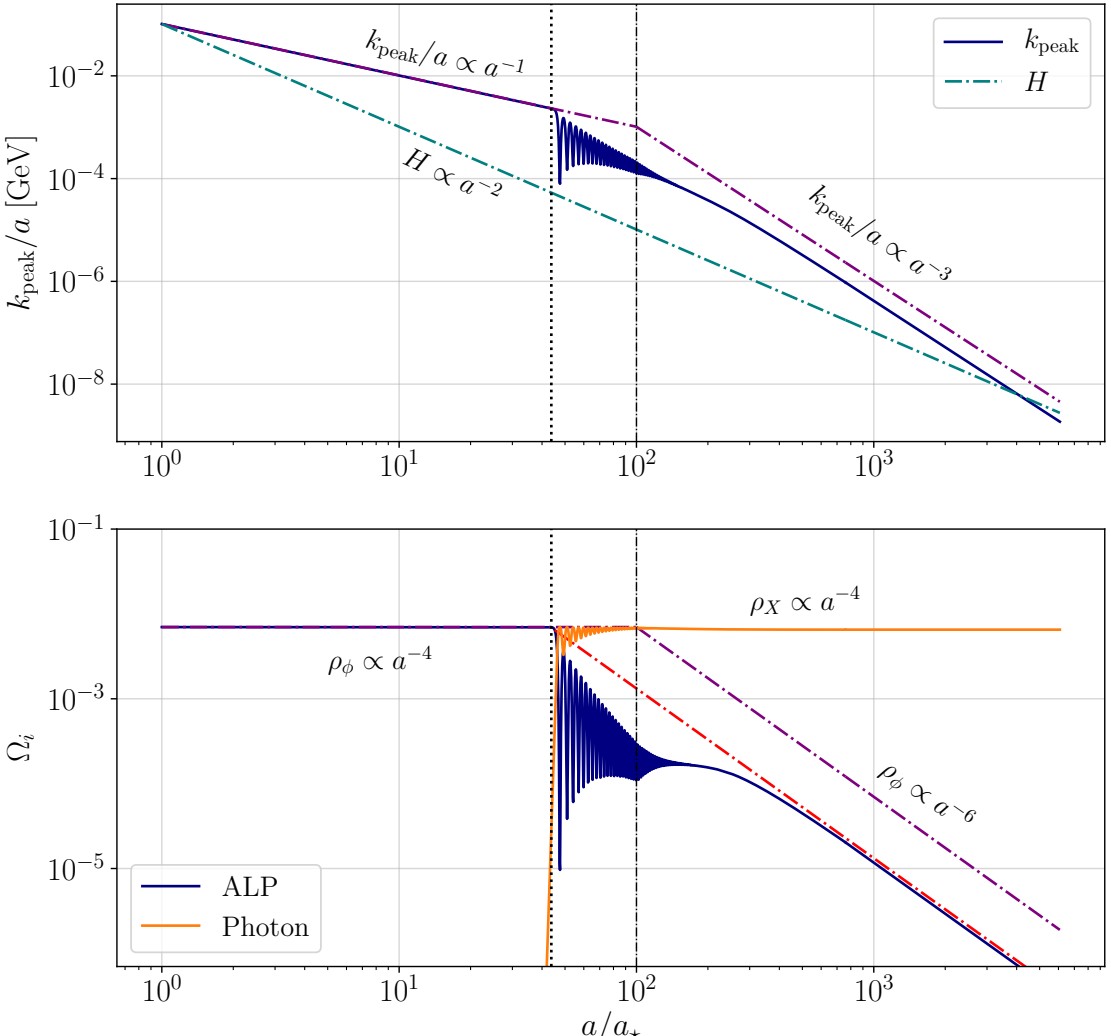

Figure 2: Comparison of rates and energy densities between a numerical simulation and analytic scaling relations for $S_i = 2 \times 10^{18}$ GeV, $m_{S,0} = 1$ GeV, $f_\phi = 5 \times 10^{13}$ GeV, $\alpha = 0.02$ and $\epsilon = 1$. We start the simulation when the Hubble rate coincides with the dark photon growth rate $k_{\mathrm{peak}}/a$ at $a = a_*$. In the top panel we show the growth rate in dark blue dominating over the Hubble rate. At $a_{\mathrm{GW}}/a_* \approx 43$ marked by the black dotted line, the growth rate deviates from the analytic scaling behavior shown as the purple, dash-dotted line. The reason for this discrepancy can be found in the bottom panel, where we can see the dark photon energy becoming comparable to the one of the axion around this time. Friction from dark photon production becomes efficient and the growth rate, which is proportional to the axions velocity, decreases faster as by the scaling only considering Hubble friction. The dash-dotted black line marks the saxion field settling at its VEV $f_\phi$ at $a_{S=f_\phi}$. Afterwards the photon production quickly becomes inefficient and all quantities take on their respective scaling behaviors, although with the growth rate and axion energy reduced due to friction from the photons. The relic ALP abundance after dark photon production is well matched by the red dash-dotted line, which denotes a kination-like scaling starting at $a_{\mathrm{GW}}$. Since we observe this behavior throughout all our simulations, we will use this as an analytic estimate of the minimum relic ALP abundance in Sec. 5.1.

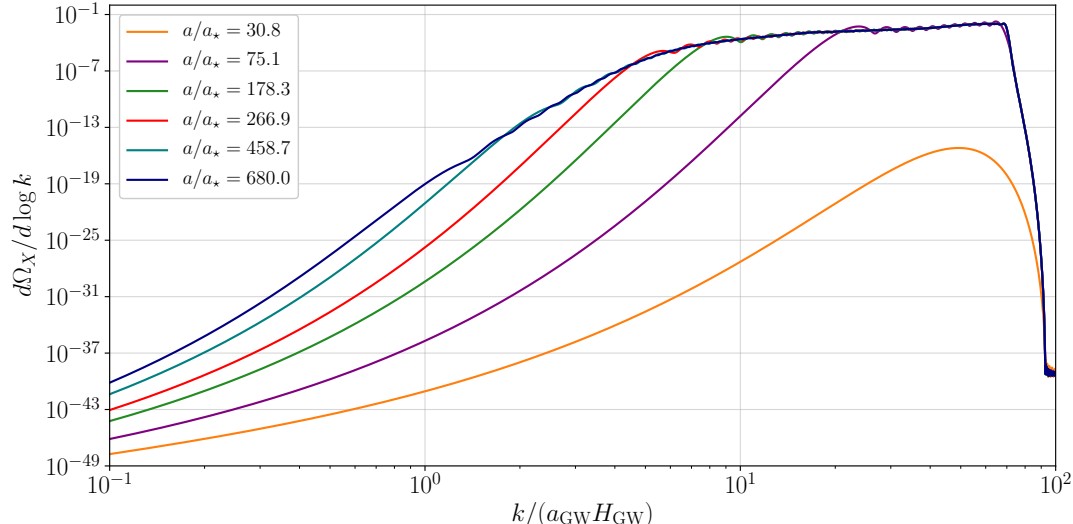

Figure 3: Evolution of the dark photon spectrum for the same simulation as in Fig. 2. The first benchmark is taken before $a_{GW} \approx 43 a_*$ and we can clearly see the peak at $k_{peak}/(a_{GW} H_{GW}) = a_{GW}/a_* \approx 43$. The next point is after the back-reaction onto the axion has become sizeable. The spectrum becomes a power-law with an exponential cutoff slightly above $k_{peak}$, as only modes with smaller momenta are amplified, due to the friction of photon production reducing the axion velocity. The other spectra are taken once the saxion has settled at its VEV and kination scaling has set in. The fast dilution of the axion velocity during this phase due to Hubble friction leads to modes at small momenta never being produced efficiently, which becomes clear from the drastic suppression of the spectrum at low momenta. Note that for this simulation we chose the axion's velocity to be positive, which is why the spectrum is fully "+"-polarized.

## 3.3 Simulation

For our numerical analysis we closely follow Ref. [2]. We solve the coupled system of $10^5$ dark photon modes with linearly spaced momenta in the range $0 < k \leq \alpha \phi'_*/S_*$, which corresponds to the tachyonic window at the start of the simulation at $a = a_*$, and the homogeneous axion field $\phi$. For the saxion field we assume however, that it follows the analytic scaling. That is

$$S(\tau) = \begin{cases} \frac{a_*}{a(\tau)} S_* & a(\tau) < a_{S=f_\phi}, \\ f_\phi & a(\tau) \geq a_{S=f_\phi}. \end{cases} \tag{36}$$

The axion EOM then takes the form

$$\phi'' + n(\tau) a H \phi' + a^2 \frac{\partial V}{\partial \phi} = -\frac{\alpha}{S(\tau)} a^2 \langle 0|\mathbf{E}\cdot\mathbf{B}|0\rangle, \tag{37}$$

where we take

$$n(\tau) = \begin{cases} 1 & \text{if } a(\tau) < a_{S=f_\phi}, \\ 2 & \text{if } a(\tau) \geq a_{S=f_\phi}, \end{cases} \tag{38}$$

in order to account for the effect of the radial saxion mode on the circular motion before the saxion settles. Note further that, for the back-reaction of the photon modes on the axion, we have to take the expectation value, since we are only considering the motion of the homogeneous axion field, $\mathbf{E}\cdot\mathbf{B} \to \langle 0|\mathbf{E}\cdot\mathbf{B}|0\rangle$. As shown in Refs. [2, 44], this expectation value can

be expressed as an integral including the mode functions, which in our simulation is approximated as[3]

$$\langle 0|\mathbf{E}\cdot\mathbf{B}|0\rangle = \frac{1}{2\pi^2 a^4}\sum_{\lambda=\pm}\sum_{k}\Delta k\, k^3\, \mathrm{Re}\left(v_\lambda(k,\tau)v_\lambda'^*(k,\tau)\right). \tag{39}$$

Let us briefly comment on the two main assumptions that we made in our analysis. The first is that the motion of the saxion field $S$ can be treated independently. Imagine for simplicity a scenario where the full field $P = S/\sqrt{2}\,\exp(i\phi/S)$ is on a circular track with the radius $S$ slowly changing due to Hubble friction, initially. Even in this idealized scenario the field would leave this circular track once friction from photon production diminishes the rotational momentum. However the main characteristics of the photon and GW spectrum would be set before these effects become sizeable. Furthermore, once the saxion $S$ starts varying rapidly, the effective field theory (EFT) in which the degrees of freedom leading to the coupling between the axion and photons, presumably fermions, have been integrated out is not valid anymore. At this point, also our semi-classical treatment of the system breaks down and, to our knowledge, there is no method to treat such a system so far. We therefore stick to the pragmatic approach here of calculating what we can, knowing that the main observables, the features of the GW spectrum, will be estimated correctly.

The second assumption is that the axion stays homogeneous throughout the evolution. Since the dark electric and magnetic fields are inhomogeneous, as becomes clear from their non-vanishing momenta in Fourier space, the backreaction onto the axion will introduce inhomogeneities. The effects of this deviation from our assumption have been studied in detail in Refs. [4,45,46,70,71]. The results therein can be summarized in that the peak of the dark photon and GW spectrum shift to slightly higher momenta, and that their polarization is washed out if the coupling of the dark photon to the axion is sufficiently strong, such that there is a prolonged period of interaction. The main deviation though occurs in the relic abundance of the axion, that cannot be reliably estimated in our analysis. In Sec. 5.1 we therefore only give a lower and upper bound.

# 4 Gravitational Wave Spectra

The exponential growth of the tachyonically produced dark photon modes leads to large anisotropies in the energy-momentum tensor. These act as a source of GWs, producing a SGWB that may be observed in current or future GW observatories. This section provides an estimate of the GW peak frequency and amplitude, as well as the spectra resulting from numerical calculations.

## 4.1 Analytic Estimates

The spectrum of a SGWB is characterized in terms of the energy density in GWs per logarithmic frequency interval normalized to the total energy density of the Universe,

$$\Omega_{\mathrm{GW}}(f) \equiv \frac{1}{\rho_{\mathrm{tot}}}\frac{d\rho_{\mathrm{GW}}}{d\log f}, \qquad \rho_{\mathrm{GW}} = \frac{M_{\mathrm{Pl}}^2}{4}\langle \dot{h}_{ij}\dot{h}^{ij}\rangle, \tag{40}$$

where $h_{ij}$ is the GW metric perturbation in transverse-traceless gauge and dots indicate derivatives with respect to cosmic time $t$. The former, as a function of conformal time $\tau$ and comoving momentum $k$, satisfy the linearized Einstein equation, which, assuming radiation

---

[3]This expression is UV divergent and in our simulation regulated by a cutoff, since we only include high enough momenta to cover the tachyonic band. Due to the exponential growth of the modes in the resonance band the result is independent of the exact value of the cutoff.

domination, read

$$\left(\partial_\tau^2 + k^2\right) a(\tau) h_{ij}(\mathbf{k}, \tau) = \frac{2 a(\tau)}{M_{\mathrm{Pl}}^2} \Pi_{ij}(\mathbf{k}, \tau).$$  (41)

Here, $\Pi_{ij}(\mathbf{k}, \tau)$ is the anisotropic stress tensor that sources the GWs, which, in the case at hand, is given by the anisotropic part of the dark photon energy-momentum tensor,

$$\Pi_{ij}(\mathbf{k}, \tau) = -\frac{\Lambda_{ij}^{ab}(\mathbf{k})}{a^2(\tau)} \int \frac{d^3q}{(2\pi)^3} \left[E_a(\mathbf{q}, \tau) E_b(\mathbf{k} - \mathbf{q}, \tau) + B_a(\mathbf{q}, \tau) B_b(\mathbf{k} - \mathbf{q}, \tau)\right],$$  (42)

where $\mathbf{E}(\mathbf{k}, \tau)$ and $\mathbf{B}(\mathbf{k}, \tau)$ are the dark electric and magnetic fields, and $\Lambda_{ij}^{ab}(\mathbf{k})$ is the transverse and traceless projector (see, e.g., Ref. [1] for further details).

As discussed in Sec. 3.2, GW emission occurs at the time $\tau_{\mathrm{GW}}$, which is delayed from the time $\tau_*$, at which the tachyonic production starts, by the growth time of the dark photon modes. Assuming that the SGWB is generated instantaneously at $\tau_{\mathrm{GW}}$, a simple estimate of the peak position of the resulting GW spectrum can be obtained as follows. Since there are two dark photon modes ($\mathbf{q}$ and $\mathbf{k} - \mathbf{q}$) contributing to the source in Eq. (42), the GW peak is obtained when both contributions come from the peak of the dark photon spectrum. The co-moving momentum at the GW peak is hence approximately given by twice the dark photon peak momentum at the time of GW production, so that we obtain the physical GW peak momentum $\tilde{k}_{\mathrm{GW}}^{\mathrm{phys.}}$ at emission,

$$\tilde{k}_{\mathrm{GW}}^{\mathrm{phys.}} = \frac{\tilde{k}_{\mathrm{GW}}}{a_{\mathrm{GW}}} = 2 \frac{k_{\mathrm{peak}}}{a_*} \frac{a_*}{a_{\mathrm{GW}}} = \frac{\alpha^2 \epsilon^2}{32\sqrt{6}} \frac{m_{S,0} S_i}{f_\phi} \left(\frac{g_{\epsilon,S_i}}{g_{\epsilon,\mathrm{GW}}}\right)^{\frac{1}{2}} \left(\frac{g_{s,\mathrm{GW}}}{g_{s,S_i}}\right)^{\frac{2}{3}} \frac{\tau_*}{\tau_{\mathrm{GW}}},$$  (43)

where we assumed that GW emission occurs before the saxion reaches its minimum, i.e. $S > f_\phi$, so that $k_{\mathrm{peak}}/a_*$ is given by Eq. (28). Note that $S_i \sim M_{\mathrm{Pl}}$ is required to obtain a sufficiently large angular velocity to generate an observable GW signal, whereas occurrence of tachyonic production constrains $\alpha/f_\phi$. Hence, the peak momentum is predominantly set by the saxion mass parameter $m_{S,0}$.

As both, the GW and dark photon spectrum, are sharply peaked, the peak amplitude can be estimated from the total GW energy density. Taking $h_{ij} \sim a^2 \Pi_{ij}/k^2$ with $\Pi_{ij} \sim \rho_\phi$, Eq. (40) can be rewritten as [2, 72, 73]

$$\tilde{\Omega}_{\mathrm{GW}} = c_{\mathrm{eff}}^2 \Omega_{\phi,\mathrm{GW}}^2 \left(\frac{H_{\mathrm{GW}}}{\tilde{k}_{\mathrm{GW}}^{\mathrm{phys.}}}\right)^2 = c_{\mathrm{eff}}^2 \left(\frac{\epsilon S_i}{M_{\mathrm{Pl}}}\right)^4 \left(\frac{a_*}{a_{\mathrm{GW}}}\right)^2 \frac{g_{\epsilon,S_i}^2}{g_{\epsilon,\mathrm{GW}} g_{\epsilon,*}} \left(\frac{g_{s,\mathrm{GW}} g_{s,*}}{g_{s,S_i}^2}\right)^{\frac{4}{3}},$$  (44)

where the factor $c_{\mathrm{eff}}$ accounts for the efficiency of converting the energy initially stored in the axion into GWs. The second part of the equation is obtained using that efficient dark photon production starts when the tachyonic window opens at $a_* H_* = k_{\mathrm{peak}}$, as well as that the axion energy density $\rho_\phi = \frac{1}{2}\dot{\phi}^2$ red-shifts as radiation for $S > f_\phi$. We have absorbed numerical factors into the efficiency factor, and $a_*/a_{\mathrm{GW}}$ is given by Eq. (34).

Finally, the present-day peak frequency and amplitude are simply obtained by red-shifting Eqs. (43) and (44),

$$\tilde{f}_{\mathrm{GW},0} = \frac{\tilde{k}_{\mathrm{GW}}^{\mathrm{phys.}}}{2\pi} \frac{a_{\mathrm{GW}}}{a_0} = \frac{\epsilon \alpha T_0}{8\pi\sqrt{6}} \left(\frac{g_{\epsilon,S_i} \pi^2}{15}\right)^{\frac{1}{4}} \left(\frac{g_{s,\mathrm{MR}}}{g_{s,S_i}}\right)^{\frac{1}{3}} \sqrt{\frac{m_{S,0}}{f_\phi} \frac{S_i}{M_{\mathrm{Pl}}}},$$  (45a)

$$\tilde{\Omega}_{\mathrm{GW},0} = \tilde{\Omega}_{\mathrm{GW}} \left(\frac{a_{\mathrm{GW}}}{a_0}\right)^4 \left(\frac{H_{\mathrm{GW}}}{H_0}\right)^2 = c_{\mathrm{eff}}^2 \frac{T_0^4}{3 M_{\mathrm{Pl}}^2 H_0^2} \frac{\pi^2 g_{\epsilon,S_i}^2}{30 g_{\epsilon,\mathrm{GW}}} \frac{g_{s,\mathrm{GW}}^{4/3} g_{s,\mathrm{MR}}^{4/3}}{g_{s,S_i}^{8/3}} \left(\frac{\tau_*}{\tau_{\mathrm{GW}}}\right)^2 \left(\frac{\epsilon S_i}{M_{\mathrm{Pl}}}\right)^4,$$  (45b)

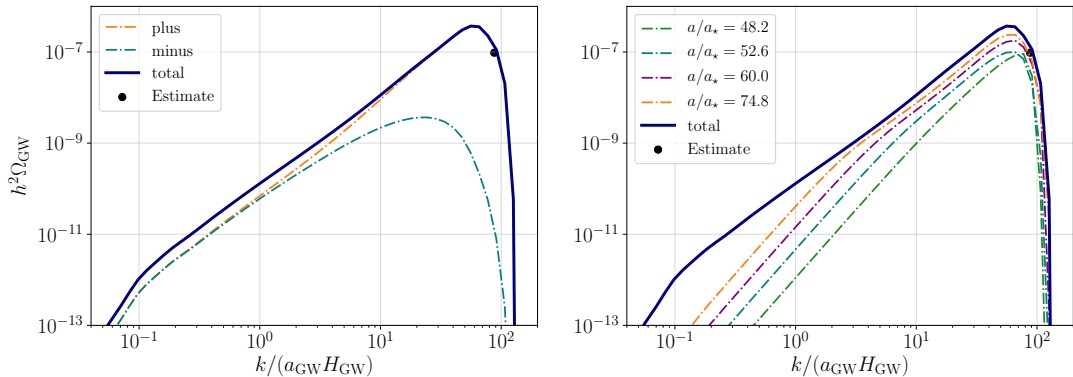

Figure 4: GW spectrum as a function of the physical momentum $k/a_{\mathrm{GW}}$ normalized to the Hubble rate at emission. The spectrum is calculated numerically from the simulated dark photon spectrum shown in Fig. 3. The parameters correspond to benchmark 2 in Table 1. The black dots indicates the estimate in Eq. (45). **Left:** GW spectrum at the time of emission. The solid blue curve depicts the total spectrum, whereas the dot-dashed orange and cyan curves correspond to positive and negative GW helicities, respectively. **Right:** Total GW spectrum at different times.

where $g_{s,\mathrm{MR}}$ are the entropic degrees of freedom at matter-radiation (MR) equality, and $T_0$ is the present-day photon temperature. Note that, fixing $\epsilon = 1$ at its maximal value, the energy budget, and, correspondingly, the peak amplitude, are predominantly determined by the initial saxion field value $S_i$, with only a weak dependence on other model parameters through the effective degrees of freedom and the growth time delay $\tau_{\mathrm{GW}}/\tau_*$.

Similar to the audible axion scenario [2,3], we expect the spectrum to drop sharply above the peak, as the generation of GWs with momenta $|\mathbf{k}| > \tilde{k}_{\mathrm{GW}}$ requires that also the contributing dark photon modes have momenta $|\mathbf{q}|, |\mathbf{k} - \mathbf{q}| > k_{\mathrm{peak}}$. Furthermore, as only one dark photon polarization is produced, the GW spectrum will also be chiral. In the low frequency tail, on the other hand, for frequencies corresponding to momenta on super-horizon scales at production, the spectrum should behave as $f^3$ based on causality arguments [1,74] (cf. also Ref. [61]).

## 4.2 Numerical Calculation

To corroborate the analytic estimates from Sec. 4.1, and to further obtain the spectral shape of the generated SGWB, the GW spectrum is also calculated numerically from the simulation discussed in Sec. 3.3, using a subset of 200 of the $10^5$ simulated modes. The computation follows the procedure in Ref. [2].

The left panel of Fig. 4 shows the resulting GW spectrum at the time of emission corresponding to the dark photon spectrum shown in Fig. 3. The solid line depicts the total spectrum, whereas the dot-dashed lines indicate the respective contributions from the positive and negative GW helicity. As expected, the resulting spectrum is dominated by the positive helicity contribution, corresponding to the sign of the initial axion velocity. The GW helicity picks up contributions from the photon helicity as well as orbital angular momentum, which is why the spectrum is only partially polarized as described in Ref. [2]. The analytic estimate Eq. (45) provides a good approximation of the peak position, assuming an efficiency factor of $c_{\mathrm{eff}} = 1$. The right panel of the figure illustrates the time dependence of the GW spectrum, showing the spectrum at different stages during the simulation. As can be seen from the figure, efficient GW emission occurs early on, with comparably little additional GWs emitted at later times. The GW peak is already pronounced at $a/a_* = 48.2$, which agrees well with the estimate of the GW emission time $a_{\mathrm{GW}}/a_* \sim 43$ obtained from Eq. (34).

Table 1: Parameter values of the benchmark spectra depicted in Fig. 5.

| Benchmark | $m_{S,0}$ [GeV] | $f_\phi$ [GeV] | $\alpha$ | $S_i$ [GeV] |
|---|---|---|---|---|
| 1 | $10^2$ | $1 \times 10^{13}$ | $4 \times 10^{-3}$ | $2 \times 10^{18}$ |
| 2 | 1 | $5 \times 10^{13}$ | $2 \times 10^{-2}$ | $2 \times 10^{18}$ |
| 3 | $10^{-2}$ | $1 \times 10^{13}$ | $4 \times 10^{-3}$ | $2 \times 10^{18}$ |
| 4 | $10^{-6}$ | $3 \times 10^{13}$ | $4 \times 10^{-2}$ | $4 \times 10^{17}$ |
| 5 | $10^{-8}$ | $1 \times 10^{13}$ | $5 \times 10^{-3}$ | $2 \times 10^{18}$ |
| 6 | $10^{-19}$ | $2 \times 10^{13}$ | $1 \times 10^{-2}$ | $2.4 \times 10^{18}$ |

Figure 5 illustrates the parameter dependence of the generated SGWB, showing the spectra for the benchmark points listed in Table 1 together with the respective estimate of the peak position based on Eq. (45). A good agreement between the peak positions and the analytic estimation can be observed. In addition, the figure shows the power-law integrated projected sensitivity (cf. Ref. [75]) of various future GW observatories. These indicate the detectability of the spectra, where a spectrum reaching into the colored region can be observed in the respective experiment. At low frequencies in the nHz region, pulsar timing arrays (PTAs) such as the Square Kilometre Array (SKA) observatory [76] are sensitive, intermediate mHz frequencies are covered by the space-based Laser Interferometer Space Antenna (LISA) [42, 77], whereas high frequencies in the Hz to kHz regime can be probed by next-generation ground based interferometers such as the Einstein Telescope (ET) [43]. The dHz region between LISA and ET can be covered by LISA successor experiments such as the Big Bang Observer (BBO) [78] or the DECi-hertz Interferometer Gravitational Wave Observatory (DECIGO) [79, 80] and its predecessor B-DECIGO [81], whereas the gap between PTAs and LISA can be bridged by fu-

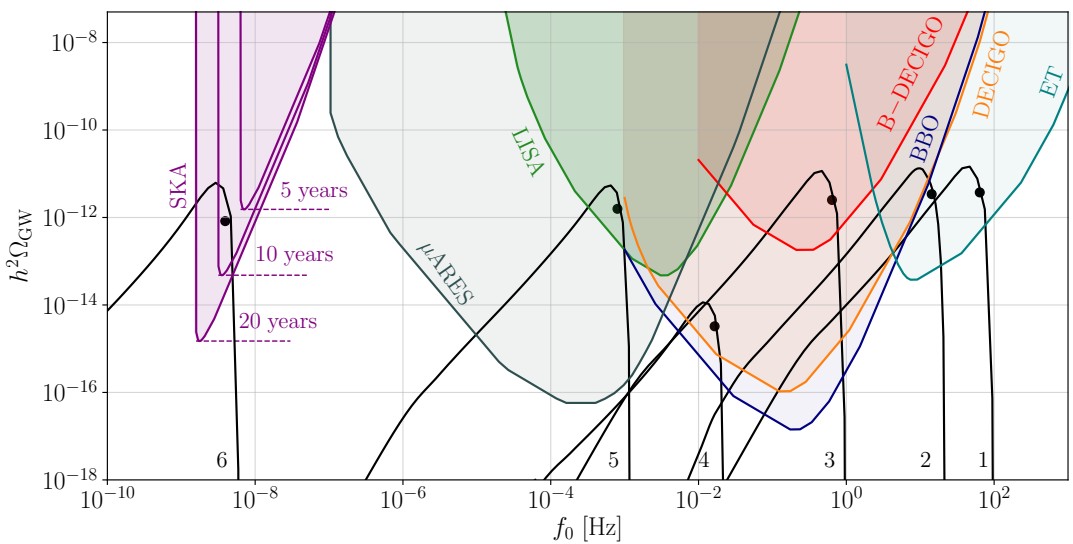

Figure 5: Present-time GW spectra (black lines) for the benchmark parameter points listed in Table 1 along with the corresponding estimates of the peak position (black dots). A good agreement between the estimates and the peak positions of the corresponding spectra obtained from simulation can be seen. The colored regions indicate the power-law integrated sensitivity of various future GW observatories. Spectra reaching into the colored regions can be detected in the respective observatory, demonstrating that, in the model considered here, observable signals can be obtained over a large range of frequencies.

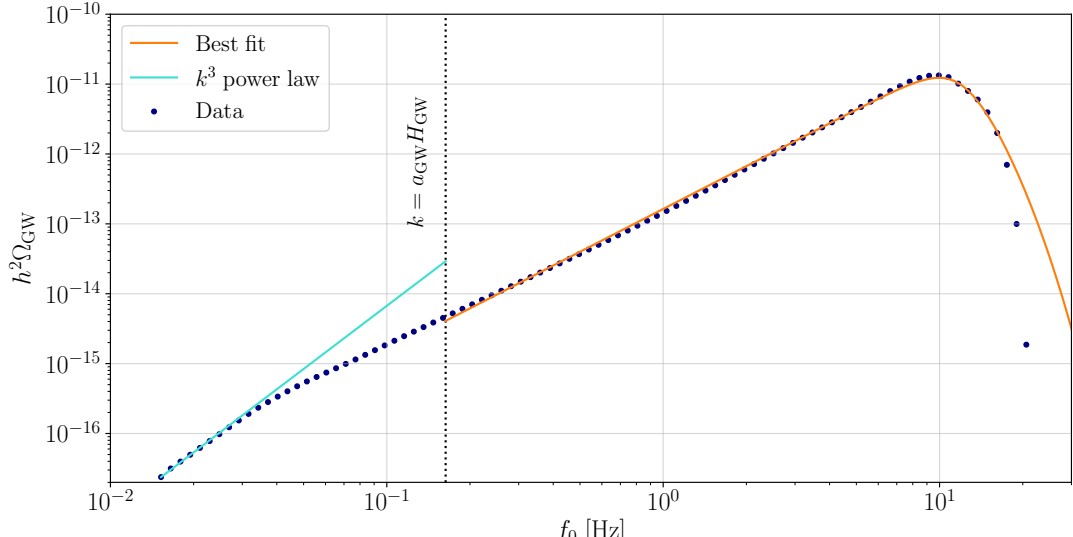

Figure 6: Simulated (blue dots) and fitted (orange line) spectrum for benchmark 2. The fit template is given by Eq. (46). At low frequencies, below the Hubble rate at the time of production indicated by the dotted vertical line, the spectrum deviates from the template, exhibiting the $f^3$ behavior indicated by the light blue line and expected from causality. When fitting the spectrum, this frequency region is hence not taken into account. The template however still captures well the spectral shape around the peak, which is the region relevant for determining the observability of the SGWB.

turistic μHz observatories such as μAres [82], which is a LISA-like mission orbiting the Sun in the orbit of Mars.[4] See Ref. [84] for further details on the assumed sensitivity curves. Within the kinetic misalignment mechanism, the axion coupled to a dark photon can produce an observable signal in any of these detectors, for suitable parameter choices with the saxion mass ranging from $m_{S,0} \gtrsim 10^{-20}$ GeV to $m_{S,0} \lesssim 10^3$ GeV.

### 4.3 Fit Template

To allow for an efficient evaluation of the detectability over the parameter space without the necessity to run the numerical simulation for every single parameter point, a fit to the spectrum based on the analytic estimates in Eq. (45) and the spectral shape template used in Ref. [3] is performed. The template takes the form

$$\Omega_{\mathrm{GW}}(f) = \mathcal{A}_s \, \tilde{\Omega}_{\mathrm{GW},0} \, \frac{\left(\tilde{f}/f_s\right)^p}{1 + \left(\tilde{f}/f_s\right)^p \exp\left[\gamma\left(\tilde{f}/f_s - 1\right)\right]} , \tag{46}$$

with $\tilde{f} = f/\tilde{f}_{\mathrm{GW},0}$, the factors $\mathcal{A}_s$ and $f_s$ correct the estimated peak amplitude and frequency, $p$ is the spectral slope below the peak, and $\gamma$ sets how sharp the spectrum is cut off above the peak.

Fitting the template to benchmark spectrum 2 from Table 1, the fit parameters are obtained as

$$\mathcal{A}_s = 6.57, \qquad f_s = 0.78, \qquad \gamma = 5.28, \qquad p = 2.04, \tag{47}$$

---

[4]There are also less futuristic ideas of bridging this gap using resonances in binaries [83], but they do not reach the sensitivities needed to constrain the model at hand.

i.e. the estimates in Eq. (45) agree with the simulations within an $\mathcal{O}(1)$ factor, and the spectrum approximately behaves as $f^2$ below the peak. The resulting fit is shown in Fig. 6. The orange line represents the fit, whereas the blue dots indicate the simulated spectrum. Only frequencies corresponding to $k > a_{\mathrm{GW}} H_{\mathrm{GW}}$ (dotted vertical line) were used in the fit, as these modes are within the horizon at the estimated time of production.

A good agreement can be observed between the simulated and fitted spectrum for frequencies below the peak. Above the peak, the simulation falls off faster than captured by the fit. It can further be seen that, in the very low frequency region, at super-horizon scales, the power of the simulated spectrum changes to $f^3$ (indicated by the light blue line), as expected from causality arguments [1, 74]. However, the region around the peak is fitted well, allowing for a sufficiently reliable determination of the detectability based on the fit.

## 4.4 NANOGrav

Recently, the NANOGrav collaboration has reported strong evidence for a stochastic common-spectrum process in their pulsar timing data [85]. Similar evidence has been found in the second data release of the Parkes Pulsar Timing Array [86], as well as by the European Pulsar Timing Array [87]. An observation of a SGWB could, however, not be claimed yet, as there is no statistical support for establishing the Hellings-Downs correlations which would be induced by a GW signal. Nonetheless, we here evaluate whether this potential SGWB may originate from rotating ALPs.

Following the procedure outlined in Ref. [88], we fit our signal template to the NANOGrav data. The best fit is obtained for a peak frequency of $f_{\mathrm{peak},0} = 6 \times 10^{-9}$ Hz and amplitude of $\Omega_{\mathrm{GW},0} h^2 \approx 2 \times 10^{-9}$. Assuming a dark-photon coupling and decay constant of $\alpha = 0.01$ and $f_\phi = 10^{13}$ GeV, respectively, this corresponds to a saxion mass of $m_{S,0} = 2.7 \times 10^{-20}$ GeV and an initial saxion field value of $S_i = 1.4 \times 10^{19}$ GeV. As the peak amplitude to first approximation only depends on $S_i$, cf. Eq. (45b), we can therefore conclude that every parameter point that is able to explain the NANOGrav signal exhibits a transplanckian value for the initial value of the saxion field. In this case the saxion would drive a period of inflation which is not taken into account by our analysis. Note furthermore that, even if one overcomes this obstacle, the dark photons sourcing the GWs would violate the current $\Delta N_{\mathrm{eff}} < 0.3$ bound according to Eq. (35), which is also a concern for the original audible axion scenario [71, 88, 89].

## 5 Relic Abundances

In the following, we compute the relic ALP abundance. Here, we distinguish the cases where the pseudoscalar represents a ALP or the QCD axion itself, respectively. In addition, we consider the possibility for the dark vectors to constitute dark matter and the conditions for successful axiogenesis.

### 5.1 ALP Dark Matter

As discussed in Sec. 2.4, assuming a slight tilt in the potential of the complex scalar $P$, the potential effectively behaves as a cosine potential for the axion, cf. Eq. (2), once its kinetic energy has been diluted and becomes comparable to the height of the potential barriers, $\dot{\phi}/2 = 2m_\phi^2 f_\phi^2$. The circular motion of the ALP then ends, and it starts to oscillate around the minimum of the cosine potential. Its energy density subsequently scales like matter, rendering the kinetically misaligned axion a candidate for DM, similar to the standard misalignment case.

In order to precisely compute the ALP relic abundance, knowledge of the energy backtransfer from the dark bosons to the ALP is crucial. These non-linear effects may be incorporated by a lattice study, as conducted in Ref. [4] for the original audible axion mechanism. However, this analysis is left for future work. Here, we limit ourselves to the estimation of a maximum and suppressed abundance scenario.

**Maximum abundance:** In this case, we ignore the energy transfer from the ALP to the dark photon. The relic abundance is then exclusively determined by the dynamics described in Sec. 2.4. The ALP kinetic energy scales kination-like between the time the saxion settles at its minimum to the start of the oscillations when $\dot{\phi}_{\text{osc}}^2/2 = 2\,m_\phi^2 f_\phi^2$, hence,

$$\left(\frac{a_{S=f_\phi}}{a_{\text{osc}}}\right)^3 = \frac{2\,m_\phi f_\phi}{\dot{\phi}_{S_i}}\left(\frac{a_{S=f_\phi}}{a_{S_i}}\right)^2. \tag{48}$$

The maximal fractional energy density of the ALP at MR equality thus becomes

$$\Omega_{\phi,\text{MR,max}} = \frac{\rho_{\phi,S_i}}{\rho_{\text{rad,MR}}}\left(\frac{a_{S_i}}{a_{S=f_\phi}}\right)^4\left(\frac{a_{S=f_\phi}}{a_{\text{osc}}}\right)^6\left(\frac{a_{\text{osc}}}{a_{\text{MR}}}\right)^3 = \frac{m_\phi f_\phi \dot{\phi}_{S_i}}{\rho_{\text{rad,MR}}}\frac{a_{S=f_\phi}}{a_{S_i}}\left(\frac{a_{S_i}}{a_{\text{MR}}}\right)^3$$
$$\simeq 0.23\,g_{\epsilon,S_i}^{-1/4}\,\frac{g_{s,\text{MR}}}{g_{\epsilon,\text{MR}}}\,\epsilon\left(\frac{S_i}{M_{\text{Pl}}}\right)^{3/2}\left(\frac{f_\phi}{m_{S,0}}\right)^{1/2}\frac{m_\phi}{T_{\text{MR}}}, \tag{49}$$

where we make the approximation $g_{\epsilon,S_i} = g_{s,S_i}$, which is a valid assumption in all of the considered parameter space.

**Suppressed abundance:** In reality, a fraction of the ALP energy density is transferred to the dark gauge field such that the final relic abundance is suppressed. As observed in our simulations (cf. Fig. 2), the relic ALP abundance after the tachyonic phase may be well approximated by an earlier onset of the kination-like scaling behavior at $a_{\text{GW}}$ instead of $a_{S=f_\phi}$, and correspondingly an earlier start of the axion oscillations. Our estimate for the suppressed abundance is hence obtained replacing $a_{S=f_\phi}$ in Eq. (48) and the first line of Eq. (49) by $a_{\text{GW}}$. Therefore,

$$\Omega_{\phi,\text{MR,min}} = \Omega_{\phi,\text{MR,max}}\frac{a_{\text{GW}}}{a_{S=f_\phi}} = \Omega_{\phi,\text{MR,max}}\frac{f_\phi}{S_i}\frac{a_*}{a_{S_i}}\frac{a_{\text{GW}}}{a_*}, \tag{50}$$

where the last two terms are given in Eqs. (30) and (34).

## 5.2 QCD Axion and Axiogenesis

If we take the ALP to be the QCD axion, we have to keep in mind that the zero temperature mass $m_{\phi,0}$ and decay constant $f_\phi$ are no longer independent parameters, but related by the QCD topological susceptibility [90] via

$$m_{\phi,0} = \frac{(78\,\text{MeV})^2}{f_\phi}. \tag{51}$$

In addition, the interaction with the thermal bath induces a suppression of the axion potential, potentially delaying the onset of oscillations. If the temperature at which axion oscillations start is smaller than the QCD scale, $T_{\text{osc}} < T_{\text{QCD}} \approx 200\,\text{MeV}$, the discussion from the previous section applies. However, if $T_{\text{osc}} > T_{\text{QCD}}$, the QCD axion undergoes an extended phase of kination-like scaling compared to the ALP from Sec. 5.1, since oscillations are then delayed until $T_{\text{QCD}}$. As the kinetic energy then is negligible comparable to the height of potential

barriers, the energy density in the ALP field at $T_{\text{QCD}}$ is determined by the potential energy. Hence, in this scenario the relic axion abundance is the same as obtained from the conventional misalignment mechanism [91].

As previously pointed out [47], the rotating QCD axion provides the possibility of successful electroweak baryogenesis. The PQ asymmetry stored in the rotation is transferred to the quark sector via QCD sphalerons. Subsequently, the chiral asymmetry is translated into the $B + L$ asymmetry by electroweak sphaleron transitions that become strongly suppressed at the electroweak phase transition (EWPT). Hence, the rotating axion constantly sources the baryon asymmetry which freezes in once the electroweak symmetry spontaneously breaks. Following the result from Ref. [47], the normalized baryon asymmetry induced by the rotation is given by

$$Y_B = \frac{n_B}{s} = \frac{45\, c_B}{2\, g_{s,\text{ws}}\, \pi^2} \frac{\dot{\theta}}{T}\bigg|_{T=T_{\text{ws}}}, \tag{52}$$

where $c_B \simeq 0.1 - 0.15\, c_W$, with $c_W$ being the weak anomaly coefficient. The observed baryon asymmetry reads $Y_B^{\text{obs}} = 8.7 \times 10^{-11}$ [69], which may immediately be converted to the required angular velocity

$$\dot{\theta}_{\text{ws}} = \frac{\dot{\phi}_{\text{ws}}}{S} = \frac{2 g_{s,\text{ws}} \pi^2}{45 c_B} Y_B^{\text{obs}}\, T_{\text{ws}} = 5.1 \times 10^{-6} \times \frac{0.1}{c_B}\, \text{GeV}, \tag{53}$$

at $T_{\text{ws}} \sim 130\,\text{GeV}$, which denotes the temperature where weak sphaleron transitions become ineffective. Since $T_{\text{ws}} > T_{\text{QCD}}$, the axion potential is flat when the baryon asymmetry freezes in, hence the pseudoscalar exhibits either a radiation- or kination-like scaling. Let us first consider the scenario of maximum velocity. In the case where the saxion has not taken its VEV at $\langle S \rangle = f_\phi$ yet at the time of electroweak symmetry breaking, we find

$$\dot{\theta}_{\text{ws,max},S>f_\phi} = \frac{\dot{\phi}_{S_i}}{S_i} \frac{a_{S_i}}{a_{\text{ws}}} \tag{54}$$

under the assumption that no dark photon production has occurred. If, however, $S = f_\phi$ at the time when the baryon asymmetry freezes in, the maximum axion velocity is given by

$$\dot{\theta}_{\text{ws,max},S=f_\phi} = \frac{\dot{\phi}_{S_i}}{S_i} \frac{a_{S_i}}{a_{S=f_\phi}} \left(\frac{a_{S=f_\phi}}{a_{\text{ws}}}\right)^3 = \frac{\dot{\phi}_{S_i}}{S_i} \left(\frac{a_{S=f_\phi}}{a_{S_i}}\right)^2 \left(\frac{a_{S_i}}{a_{\text{ws}}}\right)^3 \tag{55}$$

taking into account the additional kination-like scaling from $a_{S=f_\phi}$ to $a_{\text{ws}}$. In order to account for the production of dark photons, we again parameterize the decline of the axion velocity as a kination-like stage from $a_{\text{GW}}$ until $a_{S=f_\phi}$, yielding

$$\dot{\theta}_{\text{ws,min},S>f_\phi} = \frac{\dot{\phi}_{S_i}}{S_i} \frac{a_{S_i}}{a_{\text{GW}}} \left(\frac{a_{\text{GW}}}{a_{S=f_\phi}}\right)^2 = \frac{\dot{\phi}_{S_i}}{S_i} \left(\frac{a_{S_i}}{a_{S=f_\phi}}\right)^2 \frac{a_{\text{GW}}}{a_*} \frac{a_*}{a_{S_i}}, \tag{56}$$

for the case where $T_{S=f_\phi} < T_{\text{ws}}$. Note that Eq. (56) is a rather conservative approximation, since we assumed that the energy transfer from the axion to the dark photon is already completed. However, in this scenario, the electroweak sphalerons become inefficient during the phase of tachyonic instability, hence energy is still being transferred when the baryon asymmetry is frozen in.

For $T_{S=f_\phi} > T_{\text{ws}}$, dark photon and therefore GW production is terminated as the baryon asymmetry freezes in. Then, we obtain

$$\dot{\theta}_{\text{ws,min},S=f_\phi} = \frac{\dot{\phi}_{S_i}}{S_i} \frac{a_{S_i}}{a_{\text{GW}}} \left(\frac{a_{\text{GW}}}{a_{S=f_\phi}}\right)^2 \left(\frac{a_{S=f_\phi}}{a_{ws}}\right)^3 = \frac{\dot{\phi}_{S_i}}{S_i} \left(\frac{a_{S_i}}{a_{ws}}\right)^3 \frac{a_{S=f_\phi}}{a_{S_i}} \frac{a_{\text{GW}}}{a_*} \frac{a_*}{a_{S_i}} \tag{57}$$

for the minimum velocity.

We may now explore the parameter space that reproduces the observed baryon asymmetry. Since the GW amplitude mainly depends on the initial value of the radial degree of freedom, we fix $S_i = 2 \times 10^{18}$ GeV in the following to ensure the resulting signal lays within the reach of future observatories. Figure 7 depicts the compatible region in the $f_\phi - m_{S,0}$ plane which are then the decisive parameters regarding the GW peak frequency. In terms of the gauge coupling, we fix $\alpha = 10^{-7}$, since a large coupling leads to an earlier onset of dark photon production, hence a larger decrease of the axion velocity such that the generated baryon asymmetry is not sufficient. As a consequence of this small coupling, the axion decay constant is constrained to $f_\phi \lesssim 4 \times 10^8$ GeV to have successful GW emission. The straight blue lines in Fig. 7 denote where the axion rotation may source the baryon asymmetry under the assumption of maximum velocity, hence no dark photon production.

For $m_{S,0} \gtrsim 5 \times 10^{-5}$ GeV, the EWPT takes place during the kination-like phase, since $T_{S=f_\phi} > T_{\mathrm{ws}}$. Therefore the scaling behaviour changes between small and large saxion masses, considering the maximum velocity case. Regarding the minimum velocity scenario depicted by the dash-dotted line, we find that for successful axiogenesis, the baryon asymmetry needs to freeze in during GW production, hence $T_{S=f_\phi} < T_{\mathrm{ws}}$ in this case. Regarding the prospect of observation, we find that most of the parameter space where axiogenesis is viable may be probed by LISA, as the green shaded region suggests. In addition, also the future projects BBO, B-DECIGO, and DECIGO are sensitive to the large$-m_{S,0}$ region. The small$-m_{S,0}$ part of the

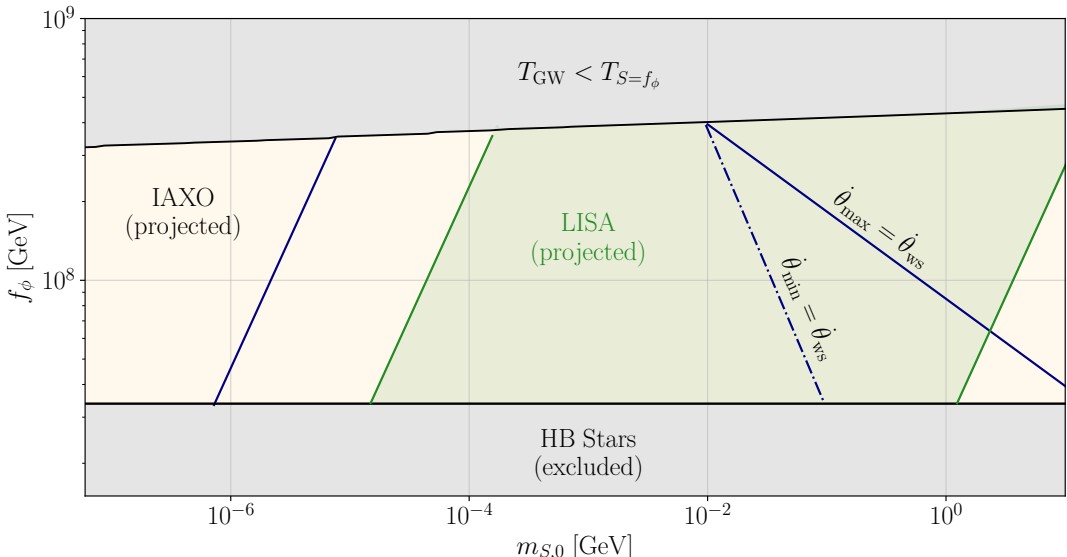

Figure 7: Parameter space of the QCD axion for for fixed benchmark parameters $S_i = 2 \times 10^{18}$ GeV and $\alpha = 10^{-7}$. On the blue lines the correct baryon symmetry is induced by weak sphaleron (WS) processes at the time of the electroweak phase transition. For large saxion masses this happens during or after the production of dark photons, where the predictability of our perturbative method is limited, which is why we show two limiting scenarios (straight and dash-dotted). In the future, the relevant parameter space may be probed by both direct axion searches such as the IAXO experiment (yellow shaded region), as well as the GW observatory LISA (green-shaded region) and potential follow-up experiments like $\mu$ARES. Note that for the chosen parameters, the relic axion abundance corresponds to the one obtained from conventional misalignment and therefore only constitutes part of DM in the shown parameter region.

shown parameter space, however, may be targeted by the projected observatory $\mu$ARES. In all of the shown parameter range the QCD potential appears so late, that the axion abundance is purely given through the normal misalignment mechanism. Assuming a misalignment angle $\theta_i = \mathcal{O}(1)$, the axion produces the correct DM abundance for $f_\phi \sim 3 \times 10^{11}$ GeV. We therefore expect the axion to only contribute a small fraction of the total DM density in the given scenario.

In order to check whether this parameter space can be tested by direct searches we relate $f_\phi$ to the coupling to the SM photon $\gamma$ through the relation $g_{a\gamma\gamma} = -1.92/(2\pi)\alpha_{\text{em}} f_\phi^{-1}$ in the case of the KSVZ axion, with $\alpha_{\text{em}} = 1/137$. We then find that Primakoff axion losses in horizontal branch (HB) stars limit $f_\phi \gtrsim 3.4 \times 10^7$ GeV [92].[5] Interestingly, the entire parameter region may be probed by the planned axion helioscope IAXO [93] for the chosen benchmark. Our model is therefore able to generate GWs over the entire range of decay constants, and we provide a multi-messenger approach for axion searches in the future, opening up parameter space that is simultaneously testable by both GWs as well as direct detection experiments.

## 5.3 Vector Dark Matter

Under the presumption that the dark photon gains a mass via a dark Stueckelberg or Higgs mechanism (cf. e.g. Refs. [3, 46]), it may in principle also constitute DM. In the following, we derive bounds for this scenario. We will simply assume that the dark vectors are massive without further specifying the spontaneous breaking of the $U(1)_X$ symmetry.

The numerical results from Sec. 3 show that the dark photon energy density is set around $a_{\text{GW}}$ and we have $\Omega_{\phi,\text{GW}} \approx \Omega_{X,\text{GW}}$, where $\Omega_{\phi,\text{GW}}$ is the axions energy density right before the emission of GWs. Furthermore, we approximate the dark photon spectrum in the following as a narrow peak around $k_{\text{peak,GW}}$. As shown in Ref. [3] the dark photon mass may not be bigger than

$$m_X \lesssim \frac{k_{\text{peak,GW}}}{a_{\text{GW}}}, \tag{58}$$

in order to not prohibit the tachyonic instability. Saturating this leads to the dark photons' energy density behaving like DM right after production, in which case it is not possible to source observable GWs without overclosing the Universe, such that the following bound is always stronger. We discuss this point for the process of axion fragmentation in Appendix A.2.

The mass will lead to the dark photon behaving like DM once the physical momentum becomes smaller than the mass at a time $a_{\text{NR}}$ given by

$$m_X = \frac{k_{\text{peak,GW}}}{a_{\text{NR}}}. \tag{59}$$

Assuming that the dark photon mass is large enough that the dark photon becomes non-relativistic before matter radiation equality, $a_{\text{NR}} < a_{\text{MR}}$, its energy density at this point is found to be

$$\Omega_{X,\text{MR}} = \Omega_{\phi,\text{GW}} \frac{a_{\text{MR}}}{a_{\text{NR}}} \frac{g_{\epsilon,\text{GW}}}{g_{\epsilon,\text{MR}}} \left(\frac{g_{s,\text{MR}}}{g_{s,\text{GW}}}\right)^{4/3}. \tag{60}$$

This allows us to set an upper bound on the temperature when the dark photon becomes non-relativistic

$$T_{\text{NR}} \leq \frac{g_{\epsilon,\text{MR}}}{g_{\epsilon,\text{GW}}} \left(\frac{g_{s,\text{GW}}}{g_{s,\text{MR}}}\right)^{4/3} \left(\frac{g_{s,\text{MR}}}{g_{s,\text{NR}}}\right)^{1/3} \frac{\Omega_{\text{DM,MR}}}{\Omega_{\phi,\text{GW}}} T_{\text{MR}}, \tag{61}$$

---

[5]Note that we do not include the potentially stronger bounds on the ALP coupling to nucleons from SN1987A, since they are less robust [92].

from the condition that the dark photon may not overclose the Universe. This bound can be restated as an upper bound on the dark photon mass.

Assuming that this bound is saturated and the dark photon is all of dark matter, one further has to take into account constraints from structure formation. Following Refs. [3, 94, 95], we set an upper bound on the boost factor, defined as the ratio of physical momentum and mass, at MR equality

$$\beta_{\mathrm{MR}} = \frac{k_{\mathrm{peak,GW}}}{a_{\mathrm{MR}} m_X} \lesssim 1.1 \times 10^{-4}, \tag{62}$$

to ensure the dark bosons are sufficiently cold at $T_{\mathrm{MR}}$ to constitute DM. Using the definition of $a_{\mathrm{NR}}$, this bound can be related to the energy density in the dark photon at emission

$$\frac{a_{\mathrm{NR}}}{a_{\mathrm{MR}}} = \frac{g_{\epsilon,\mathrm{GW}}}{g_{\epsilon,\mathrm{MR}}} \left( \frac{g_{s,\mathrm{MR}}}{g_{s,\mathrm{GW}}} \right)^{4/3} \frac{\Omega_{X,\mathrm{GW}}}{\Omega_{\mathrm{DM,MR}}} \lesssim 1.1 \times 10^{-4}. \tag{63}$$

By inserting the ALP energy density before GW production for $\Omega_{X,\mathrm{GW}}$, we find

$$\epsilon S_i \lesssim 0.05 \, M_{\mathrm{Pl}} \left( \frac{g_{\epsilon,\mathrm{MR}}}{g_{\epsilon,S_i}} \right)^{1/2} \left( \frac{g_{s,S_i}}{g_{s,\mathrm{MR}}} \right)^{2/3} \tag{64}$$

as the condition on the initial saxion value if the dark photon amounts to all of DM.

# 6 Main Results and Summary

Given the the fit template of the GW spectrum from Sec. 4.3 and the relic abundances computed in Sec. 5, we now evaluate the prospect of the presented model to both produce detectable signals as well as provide viable DM candidates.

In Fig. 8, we present the regions in the $S_i - m_{S,0}$ plane that may be probed by next-generation GW observatories. Here, we fix $f_\phi = 10^{12}$ GeV and $\alpha = 0.02$. As shown in Sec. 4, the amplitude of the GWs mainly depends on the initial value of the radial component $S_i$ which sets the energy budget available to be converted into gravitational radiation. We find that $S_i \gtrsim 10^{17}$ GeV is required for the GWs to be within the reach of future detectors. For smaller values of $S_i$, it is also not ensured that the EFT is valid, since fermions with mass $y_\psi S_i$ are produced thermally then. The frequency of the resulting GW spectrum is controlled by the gauge coupling $\alpha$, the ALP decay constant $f_\phi$ and the saxion vacuum mass $m_{S,0}$. For the chosen parameters, detectable GWs are produced between $10^{-23}$ GeV $< m_{S,0} < 10^2$ GeV, where the low masses may be probed by PTAs and the larger masses lie within the reach of future interferometers. In Sec. 5, we have computed upper and lower limits for a potentially massive dark photon to constitute DM. This bound evaluates to $S_i \lesssim 2 \times 10^{17}$ GeV, as denoted by the blue shaded region in Fig. 8. In that regime, the dark photons neither overclose the Universe at matter-radiation equality, nor violate the bounds from structure formation. Apart from $S_i$, the model parameters only impact the dark photon bound via the effective degrees of freedom at the time the saxion starts to roll. Hence, $S_i \lesssim 2 \times 10^{17}$ GeV is valid for a large range of parameters. It should be stressed that for any value of $f_\phi$, we find a probeable parameter region where the ALP is a viable DM candidate, provided $\alpha$ is large enough to allow for efficient dark photon production.

In Fig. 9, we further illustrate the dependence of the model on the respective parameters. Here, we plot the $m_\phi - m_{S,0}$ parameter plane for two different values of each $S_i$ and $f_\phi$, with the gauge coupling fixed at $\alpha = 0.1$. The blue band denotes the regime where the relic ALP abundance matches the measured DM energy density at matter-radiation equality. The straight (dash-dotted) line corresponds to the maximum (minimum) abundance case from Sec. 5.1. As

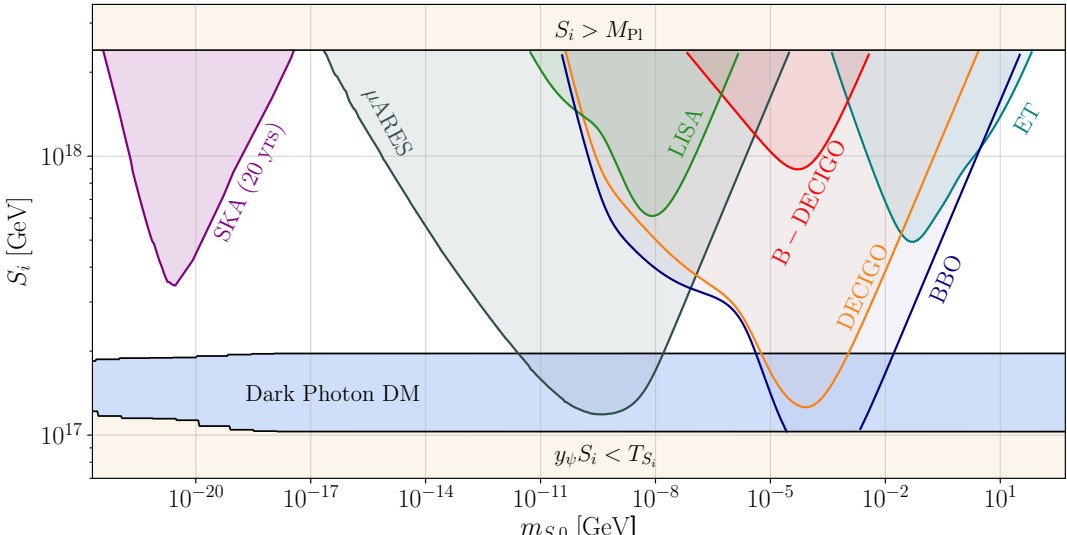

Figure 8: The parameter regions in the $S_i - m_{S,0}$ plane that are probable in the future, for the benchmark parameters $f_\phi = 10^{12}$ GeV and $\alpha = 0.02$. Since the GW amplitude is independent of $f_\phi$, the presented model provides detectable signals for any value of $f_\phi$. Hence, also the parameter region that is subject to direct axion searches may be probed via GWs in the future. In the entire parameter space, the ALP can constitute DM, since the GW production is independent of $m_\phi$. The blue shaded region marks where the dark photons, if they were massive, are sufficiently cold to be DM, but also do not overclose the Universe at $T_{MR}$.

the ALP mass $m_\phi$ is taken independent from the decay constant $f_\phi$, there exists a value of $m_\phi$ for any $m_{S,0}$, $S_i$, and $f_\phi$ that gives the correct relic abundance. The colored bands depict the sensitive regions of several future experiments. These are independent of $m_\phi$ as shown in Sec. 4.1. Hence, GW production and relic ALP abundance are decoupled, which is one of the main improvements of the finite ALP velocity. In the original audible axion model [2–4], both the GW peak frequency as well as relic ALP abundance are controlled by $m_\phi$, leading to an over-production of ALPs in large parts of the parameter space.

Furthermore, the dependency of the model on $S_i$ is shown in Fig. 9. Since $S_i$ controls the GW peak amplitude, the width of the detection bands decreases when changing the initial saxion value from $S_i = 2 \times 10^{18}$ GeV to $S_i = 5 \times 10^{17}$ GeV. By varying the ALP decay constant from $f_\phi = 5 \times 10^{13}$ GeV to $f_\phi = 5 \times 10^{11}$ GeV, however, the colored bands are shifted to smaller $m_{S,0}$, while their widths remain invariant. Hence, in contrast to the original audible axion model [2,3], the GW amplitude is independent of $f_\phi$. As a result, the low$-f_\phi$ parameter space that is subject to direct ALP searches may also be probed via GWs in the future.

Kinetic misalignment is a powerful tool for model building that allows new scenarios for axion, ALP and dark photon dark matter [61], baryogenesis [47,57] and gravitational wave phenomenology [58–60]. Combined with the audible axion mechanism [2,3], kinetically misaligned ALPs coupled to dark photons produce an observable GW signal, as already pointed out in [61]. In the present work, the dynamics leading to observable GW production are discussed in detail. Analytic estimates of the GW signal as well as a detailed numerical computation of the GW spectrum are obtained. An analytic fit to the signal is provided that allows for fast computation of signal-to-noise ratios at GW experiments, and which we use to identify the regions of parameter space that may be probed in the future.

Compared with the original audible axion scenario, kinetic misalignment renders the GW amplitude independent of $f_\phi$. It follows that a large range of decay constants now become

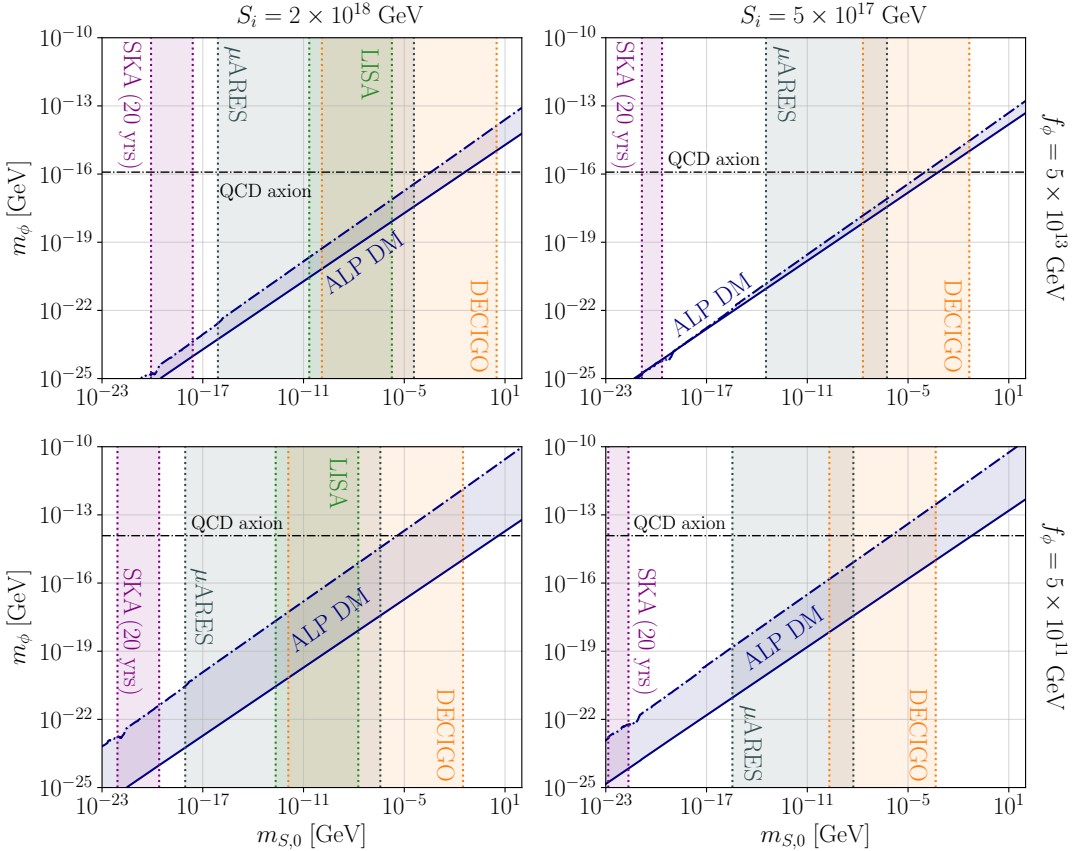

Figure 9: Parameter region in the $m_\phi - m_S$ where the ALP has the correct relic abundance to constitute DM for a benchmark coupling of $\alpha = 0.1$. The blue band represents the maximum and minimum abundance limits computed in Sec. 5.1. Here, we choose two different benchmark values for $f_\phi$ and $S_i$. The colored bands show the sensitive regions of several projected GW experiments. Clearly, the width of the detection bands does not change when altering $f_\phi$, hence detectable GWs are produced over the entire range of ALP decay constants, provided $S_i$ is sufficiently large. In addition, we find a mass configuration for every detectable signal that gives the correct relic abundance for the ALP to constitute DM.

accessible experimentally, and that the ALP DM abundance can be brought into agreement with observations easily. In parts of the parameter space, the DM can also be explained by a combination of ALPs and massive dark photons. As shown in Fig. 7, there is a region consistent with axiogenesis which may be probed by LISA and other future GW experiments.

Even in the absence of dark photons, kinetically misaligned ALPs should produce some amount of gravitational radiation due to the process of fragmentation [62]. It turns out however that there is no region of parameter space where the signal is observable experimentally and that is not already excluded by cosmological constraints. The details of our calculation and the results can nevertheless be found in the appendix.

As already mentioned, a more precise computation of the ALP DM abundance requires including the backscattering of dark photons into ALPs, which would be interesting to explore in the future, but requires a full lattice simulation. Furthermore, it would be worthwhile to explore different possible UV completions for spinning axions such as trapped misalignment [51], and to include the dynamics of the saxion field in the simulations. We leave these challenging open questions for future studies.

# Acknowledgements

We would like to thank Enrico Morgante for useful discussions and Geraldine Servant, Philip Sorensen, Cem Eroncel and Ryosuke Sato for discussions regarding their preliminary results on GWs from ALP fragmentation. Work in Mainz is supported by the Deutsche Forschungsgemeinschaft (DFG), Project No. 438947057 and by the Cluster of Excellence "Precision Physics, Fundamental Interactions, and Structure of Matter" (PRISMA+ EXC 2118/1) funded by the German Research Foundation (DFG) within the German Excellence Strategy (Project No. 39083149). The work of E.M. is supported by the Minerva Foundation.

# A ALP Fragmentation

As pointed out in previous literature [62–64], a finite ALP velocity may lead to the fragmentation of the pseudoscalar. This process allows for an energy transfer from the homogeneous zero mode to ALP excitations. Similarly to the dark photon case, a SGWB is generated, sourced by anisotropic stress from exponential particle production. In the following, we study the fragmentation process and the resulting GW production within the presented mechanism. In addition, we give the cosmological bounds constraining this scenario.

## A.1 Production of excited ALPs

Before presenting the numerical results, we analytically estimate the parameter region where fragmentation effects may become important. Below we assume that no tachyonic photon production has taken place, since otherwise the energy available to source GWs is reduced. In addition, we assume that the radial component has settled at its VEV $\langle S \rangle = f_\phi$ such that $P$ rotates around the minimum of the potential and we may regard the ALP independently. Again, we assume that the $U(1)_{\text{PQ}}$ symmetry is explicitly broken by the potential in Eq. (2). The full equation of motion describing the evolution of the axion field is

$$\ddot{\phi} + 3H\dot{\phi} - \frac{1}{a^2}\nabla^2\phi + \frac{\partial V}{\partial \phi} = 0, \tag{65}$$

where the dots denote derivatives with respect to cosmic time $t$ and the spatial derivatives account for possible inhomogeneities. Following Ref. [62], we decompose the ALP field $\phi$ as

$$\phi(\mathbf{x}, t) = \phi(t) + \delta\phi(\mathbf{x}, t) = \phi(t) + \left( \int \frac{d^3k}{(2\pi)^3} a_{\mathbf{k}} u_k(t) \exp(i\,\mathbf{k}\cdot\mathbf{x}) + \text{h.c.} \right), \tag{66}$$

with $\phi(t)$ being the homogeneous zero mode. The second term represents the excited ALP quanta, where the mode functions are given by $u_k(t)$. The respective creation and annihilation operators follow the usual commutation relation, $[a_{\mathbf{k}}, a_{\mathbf{k}'}^\dagger] = (2\pi)^3 \delta^{(3)}(\mathbf{k} - \mathbf{k}')$. The initial condition for an excited ALP mode with momentum $k$ is again taken as the Bunch-Davies vacuum, $u_k(k, \tau \to 0) = \exp(-ik\tau)/\sqrt{2k}$. This is the most conservative choice, since in principle scalar modes, in contrast to vectors that we discussed in the main text, can be excited during inflation if they leave the horizon. Recent numerical studies have however shown that the exact initial condition has only a small effect on the efficiency of the fragmentation process [64].

Further following Ref. [62], we find that the equations of motion of both the zero and

excited modes read

$$\ddot{\phi} + 3H\dot{\phi} + \frac{\partial V}{\partial \phi} + \frac{1}{2}\frac{\partial^3 V}{\partial \phi^3}\int \frac{d^3 k}{(2\pi)^3}|u_k|^2 = 0\,,$$

$$\ddot{u}_k + 3H\dot{u}_k + \left[\frac{k^2}{a^2} + \frac{\partial^2 V}{\partial \phi^2}\right]u_k = 0\,. \tag{67}$$

We are interested in the case, where, initially, the kinetic ALP energy dominates over the potential energy, i.e. $\dot{\phi}^2 \gg m_\phi^2 f_\phi^2$, since otherwise the problem reduces to the standard misalignment scenario. In order to make an analytic estimate, we neglect Hubble friction and the backreaction on the homogeneous field from the production of axion fluctuations, since it takes some time until their energy becomes comparable to the one in the homogeneous field. With those simplifications, the velocity of the homogeneous part is constant, $\phi(t) = \dot{\phi}\,t$, and the equation for the fluctuations becomes

$$\ddot{u}_k + \left[\frac{k^2}{a^2} + m_\phi^2 \cos\left(\frac{\dot{\phi}}{f_\phi}t\right)\right]u_k = 0\,. \tag{68}$$

Hence, the equation of motion describing the dynamics of the excited ALP modes takes the form of a Mathieu equation (cf. e.g. Refs. [15, 96, 97]), which has exponentially growing solutions in certain momentum bands. The instability band experiencing the fastest growth is given by

$$\left|\frac{k}{a} - \frac{\dot{\phi}}{2f_\phi}\right| < \frac{m_\phi^2 f_\phi}{2\dot{\phi}}\,, \tag{69}$$

where $k_{\rm cr}/a = \frac{\dot{\phi}}{2f_\phi}$ is the mode that grows the fastest, and $\delta k_{\rm cr}/a = \frac{m_\phi^2 f_\phi}{2\dot{\phi}}$ determines the width of the instability band as well as the typical rate at which the modes in the band grow.

## A.2 Analytic Considerations

Let us start this section with a simple argument, why fragmentation processes are not able to produce GWs observable with pulsar timing or laser interferometry without the ALPs relic density overclosing the Universe. To do so, we again use Eq. (44) to estimate the amount of produced GWs. The first factor is the amount of energy acting as a source of GWs, in our case the axion's energy. We have $\dot{\phi} \gtrsim m_\phi f_\phi$ such that the axion actually rolls over the potential barriers and, therefore, $k_{\rm cr}/(a_{\rm GW}m_\phi) \gtrsim 1$ which implies that the energy density in the excited axion modes redshifts like radiation for some time after GW emission before starting to behave like matter and contributing to DM. Maximizing the amount of energy in the axion without overclosing the Universe therefore amounts to

$$\Omega_{\phi,\rm GW} \approx \frac{1}{2}\frac{a_{\rm GW}}{a_{\rm MR}}\frac{k_{\rm cr}}{a_{\rm GW}m_\phi} = \frac{k_{\rm cr}}{2a_{\rm MR}m_\phi}\,. \tag{70}$$

The second factor of Eq. (44) includes the characteristic scale of the GW source that also sets the frequency of the waves. In the fragmentation case, it is given by $k_{\rm cr}$ and directly fixed by today's frequency $\tilde{f}_{\rm GW,0} = k_{\rm cr}/(2\pi a_0)$. In the following we are interested in the maximum GW amplitude we can produce at a frequency given by an experiment under consideration. We therefore consider $\tilde{f}_{\rm GW,0}$ to be fixed. Putting it all together we find

$$\tilde{\Omega}_{\rm GW} \approx \Omega_{\phi,\rm GW}^2\left(\frac{a_{\rm GW}H_{\rm GW}}{k_{\rm cr}}\right)^2 \approx \left(\frac{a_{\rm GW}H_{\rm GW}}{a_{\rm MR}m_\phi}\right)^2\,. \tag{71}$$

In this simplified treatment we are only left with two variables: $m_\phi$ and $a_{\mathrm{GW}}$, where the latter one fixes $H_{\mathrm{GW}}$ given the standard cosmological history. From the formula above it is clear that we want to minimize $m_\phi$, while maximizing $a_{\mathrm{GW}}H_{\mathrm{GW}}$. This ratio is however limited by a strict hierarchy of scales that is at the heart of the fragmentation process: As previously discussed, we have $k_{\mathrm{cr}}/a_{\mathrm{GW}} \gtrsim m_\phi$. Furthermore, one can easily show that $k_{\mathrm{cr}}\delta k_{\mathrm{cr}} = a_{\mathrm{GW}}^2 m_\phi^2$, and efficient production of excited axions requires the growth rate $\delta k_{\mathrm{cr}}/a_{\mathrm{GW}}$ to be bigger than the Hubble rate such that we have in total

$$H_{\mathrm{GW}} \lesssim \frac{\delta k_{\mathrm{cr}}}{a_{\mathrm{GW}}} \lesssim m_\phi \lesssim \frac{k_{\mathrm{cr}}}{a_{\mathrm{GW}}} \ . \tag{72}$$

It is therefore easy to convince oneself that the GW amplitude is maximized when this hierarchy is as small as possible, which corresponds to all of the scales above being of the same order of magnitude and the axion barely managing to roll over the barriers, $\dot\phi^2 \approx m_\phi^2 f^2$. From $k_{\mathrm{cr}} \approx a_{\mathrm{GW}}H_{\mathrm{GW}}$ we can determine $a_{\mathrm{GW}}$ and express the maximum GW amplitude as a function of solely the GW frequency today,

$$\tilde\Omega_{\mathrm{GW},0} \approx \Omega_{\mathrm{rad},0}\left(\frac{a_{\mathrm{MR}}}{a_0}\right)^2\left(\frac{H_{\mathrm{MR}}}{\tilde{f}_{\mathrm{GW},0}}\right) = 5\times 10^{-21}\left(\frac{10^{-8}\,\mathrm{Hz}}{\tilde{f}_{\mathrm{GW},0}}\right)^2 \ . \tag{73}$$

This very rough estimate agrees to within one order of magnitude with the one found in [63], although in their setup the above mentioned hierarchy was small by construction and our result can be seen as a generalization. From this estimate, it becomes clear that detection in future pulsar timing arrays like SKA with sensitivities down to $\Omega_{\mathrm{GW}} \approx 10^{-15}$, let alone laser interferometers with similar sensitivity but at higher frequencies, is not possible in a general fragmentation setup without additional suppression of the axion abundance. Below, we investigate the GWs from fragmentation and stay heuristic to how this suppression is accomplished (see e.g. Refs. [4,63] for possible mechanisms). Finally, let us note that there is in principle a more stringent bound for efficient growth than $H_{\mathrm{GW}} < \delta k_{\mathrm{cr}}/a_{\mathrm{GW}}$ since the critical momentum and, therefore, the amplified modes are red-shifting, as well as due to growth time considerations, similar to the ones in the main text, that can be found in Refs. [62,64] but were skipped here for simplicity.

Let us now get back to the specific case, where the intial velocity is generated by the saxion motion. The constraint that the field rolls over the potential barriers at first can be related to the saxion mass

$$\dot\phi_{S=f_\phi} = \epsilon m_{S,0}f_\phi \gtrsim m_\phi f_\phi. \tag{74}$$

We can furthermore relate the source energy density to the saxion parameters through

$$\Omega_{\phi,\mathrm{GW}} = \left(\frac{\epsilon S_i}{M_{\mathrm{Pl}}}\right)^2\left(\frac{a_{S=f_\phi}}{a_{\mathrm{GW}}}\right)^2 \ , \tag{75}$$

which implies that, again, we have to choose $\epsilon S_i$ close to the Planck scale. Furthermore, $a_{S=f_\phi}/a_{\mathrm{GW}}$ should not be too small. If, at the same, time we want to keep the hierarchy of scales small to get the maximum amount of GWs, this implies that the ratio between $\epsilon m_{S,0}$ and $m_\phi$ cannot be too large, which is what we consider in our simulations below.

## A.3 Numerical Results and Gravitational Waves

We start the simulation when the radial component has settled at $S = f_\phi$. This takes place at

$$T_{S=f_\phi} = T_{S_i}\frac{f_\phi}{S_i}\left(\frac{g_{s,S_i}}{g_{s,S=f_\phi}}\right)^{1/3} \ , \tag{76}$$

where $T_{S_i}$ is given by Eq. (11). The expansion rate of the Universe at that time reads

$$H_{S=f_\phi} = H_{S_i} \left( \frac{g_{\epsilon,S=f_\phi}}{g_{\epsilon,S_i}} \right)^{\frac{1}{2}} \left( \frac{g_{s,S_i}}{g_{s,S=f_\phi}} \right)^{\frac{2}{3}} \left( \frac{f_\phi}{S_i} \right)^2 = \frac{m_{S,0}}{\sqrt{6}} \frac{f_\phi}{S_i} \left( \frac{g_{\epsilon,S=f_\phi}}{g_{\epsilon,S_i}} \right)^{\frac{1}{2}} \left( \frac{g_{s,S_i}}{g_{s,S=f_\phi}} \right)^{\frac{2}{3}} . \tag{77}$$

Regarding the numerical procedure, we take a similar approach as described in Sec. 3.3. The coupled equations of motion Eq. (67) are translated to conformal time, giving

$$\phi'' + 2aH\phi' + a^2 m_\phi^2 f_\phi \sin\left( \frac{\phi}{f_\phi} \right) - \frac{1}{4\pi^2} \frac{m_\phi^2}{f_\phi} \sum_k \Delta k k^2 |u_k|^2 = 0 ,$$

$$u_k'' + 2aH u_k' + \left[ k^2 + a^2 m^2 \cos\left( \frac{\phi}{f_\phi} \right) \right] u_k = 0 . \tag{78}$$

Here, the ALP field is discretized into $N_k = 10^5$ modes within the range

$$0 < k < 2 k_{\rm cr} = \frac{\dot{\phi}_{S=f_\phi}}{f_\phi} a_{S=f_\phi} . \tag{79}$$

The initial velocity $\dot{\phi}_{S=f_\phi}$ is given by Eq. (74). We stop the simulation after the energy density of the excited modes has grown to a value comparable to the zero mode's energy density. Note that our approach treats $\delta\phi$ as a small fluctuation and therefore in principle one expects deviations when $\rho_{\delta\phi}$ grows large as perturbativity breaks down. Numerical studies of this and similar systems that were not relying on any expansion, however, found no qualitative differences, cf. e.g. Refs. [64, 98].

Figure 10 shows the results of the simulation conducted with the benchmark 2 parameters from Table 2. As the initial kinetic energy exceeds the maximum of the potential by a factor of $\mathcal{O}(10)$, the zero mode rolls over several maxima, before being trapped at the minimum depicted by the dotted line in the bottom plot. From there, oscillations around the minimum start and the homogeneous ALP mode behaves like pressureless matter. The upper plot shows the energy densities during the simulation, where the kinetic energy density of the zero mode is denoted by the orange line. The blue line represents the evolution of the excited modes. Before the effects from fragmentation become sizeable at $a \sim 1.8 \times a_{S=f_\phi}$, the total energy density of the zero mode stays conserved up to cosmic expansion. Then, the energy density of the excited modes becomes comparable to the zero mode, and we stop the simulation soon after.

Note that in this benchmark the energy in the fluctuations only become comparable after the axion is trapped in one minimum due to Hubble friction. This corresponds to the limit where the hierarchy between scales that we discussed in the previous section is not present, such that all relevant length and time scales are of the same order. From our analytic discussion we expect this scenario to maximise the GW amplitude.

The rapid growth of excited ALP quanta then leads to the production of stochastic GWs. We extract the mode functions from the simulation and compute the resulting GW spectrum. The full calculation may be found in Appendix A.4. Figure 11 shows snapshots of both the excited ALP spectrum, as well as the GW signal for the benchmark 2 parameters at different times during the evolution. We observe that the ALP spectrum grows exponentially, as expected, and reaches its final peak amplitude close to the end of the simulation. As the system evolves, the instability band, and with it the peak of the spectrum, moves to smaller momenta. This is expected, since the peak momentum is proportional to the zero mode's velocity that decreases through cosmic expansion and particle production. The dotted line corresponds to an estimate of $k_{\rm cr}$ at $a/a_{S=f_\phi} = 1.91$, when most of the excited axion quanta are produced, and can be taken as a good estimator for the peak momentum of the final axion spectrum.

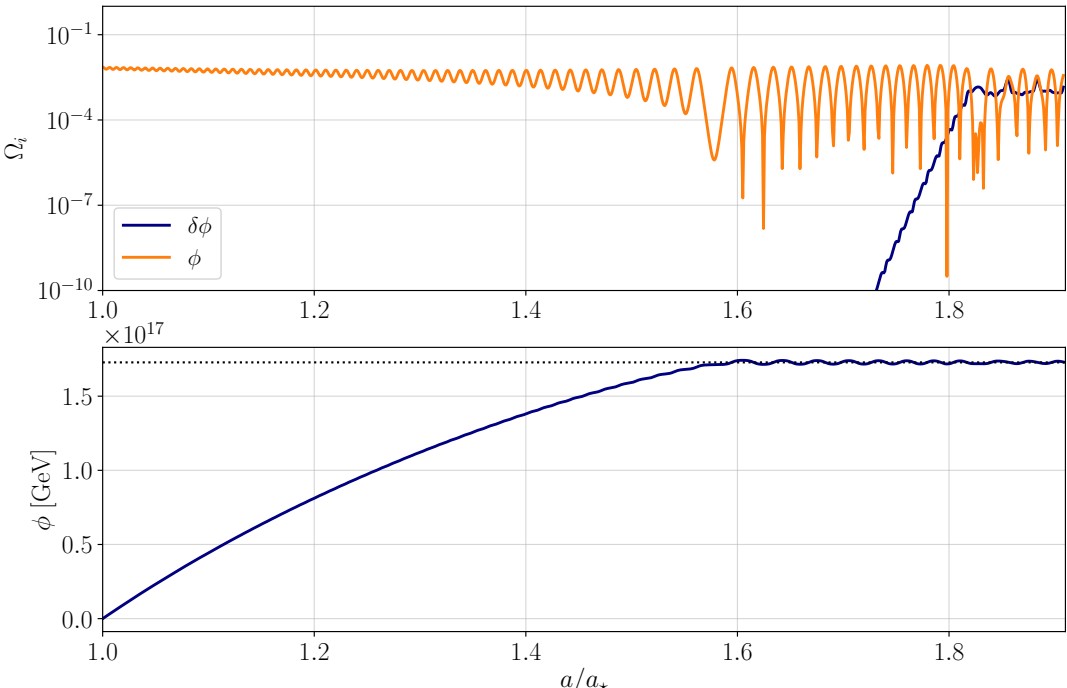

Figure 10: **Top:** The fractional energy densities of both, the zero mode as well as the excited modes. Note that, in terms of the zero mode, solely the kinetic energy density is depicted. **Bottom:** Evolution of the zero mode. The ALP rolls over the maximum several times before it gets trapped at the minimum indicated by the black dotted line.

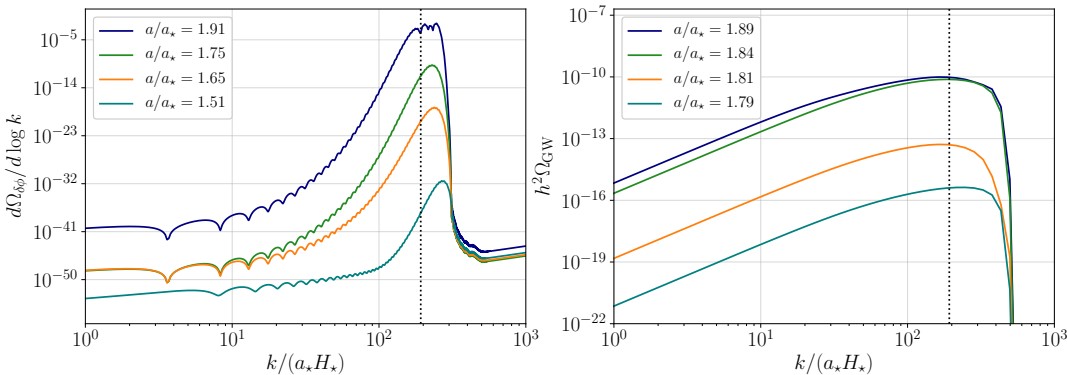

Figure 11: The spectrum of the excited ALP modes (left) and the resulting GWs (right), normalized to the total energy density at the start of the simulation. The coloured curves correspond to snapshots of the spectrum at different times. The dotted black line denotes the analytic estimate of the peak momentum $k_{\mathrm{cr}}$ at $a/a_* = 1.91$.

In terms of the GW spectrum, it is observed that the peak builds up during the very end of the simulation, when most of the energy has been transferred into the excited ALP modes. The simulation matches our expectation that the peak of the GW spectrum should be close to $k_{\mathrm{cr}}$, as one can see on the right side of Fig. 11.

Figure 12 shows the results of our simulation conducted with different benchmark parameters that may be found in Table 2. Similarly to the case with dark photon production, a SGWB may be generated over a large frequency range by altering the saxion vacuum mass.

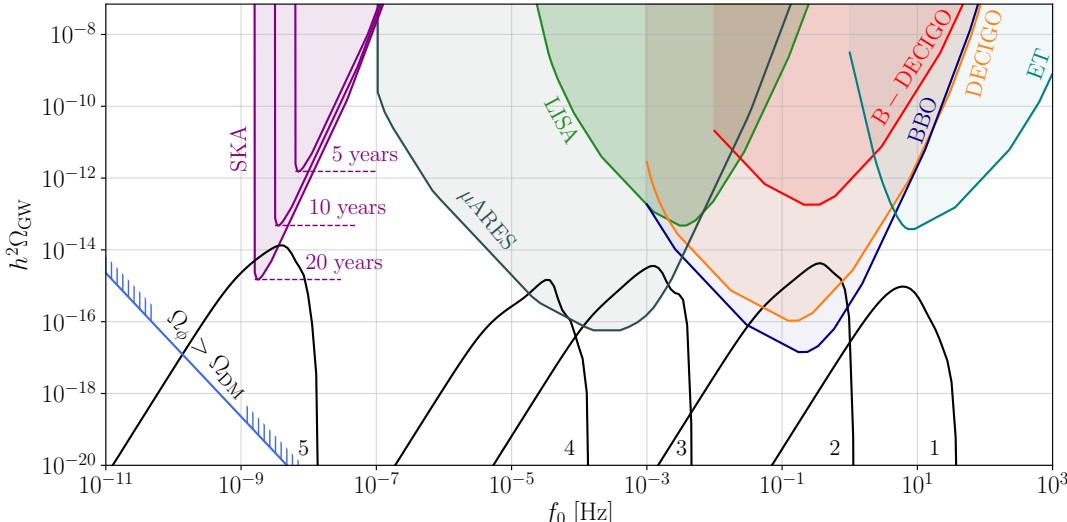

Figure 12: Today's gravitational wave spectra, calculated with the benchmark parameters given in Table 2. The coloured regions correspond to the power-law integrated sensitivities of the respective observatories. The blue line corresponds to the estimate from Eq. (73). We find that the axion overcloses the Universe in the entire detectable parameter space.

As discussed in the previous section, the axion mass is then bound to be one to two orders of magnitude smaller than $m_{S,0}$ to produce observable GWs.

The initial saxion value $S_i$ again sets the initial kinetic energy of the ALP zero mode, and is hence fixed to lie in the vicinity of $M_{\mathrm{Pl}}$ to have a sufficiently large energy budget in the first place. The impact of the ALP decay constant $f_\phi$ is a bit more subtle and may be seen when regarding the scale of GW emission. By combining Eq. (77) and Eq. (79), we find

$$\frac{k_{\mathrm{cr},S=f_\phi}}{a_{S=f_\phi} H_{S=f_\phi}} = \frac{\epsilon}{8} \frac{S_i}{f_\phi} \left( \frac{g_{\epsilon,S_i}}{g_{\epsilon,S=f_\phi}} \right)^{1/2} \left( \frac{g_{s,S=f_\phi}}{g_{\epsilon,S_i}} \right)^{2/3} . \tag{80}$$

Hence, for small $f_\phi$ the ratio $k_{\mathrm{cr}}/aH$ grows, which corresponds to a suppression of the GW peak amplitude. As $f_\phi$ approaches $S_i$, the GW peak momentum moves closer to the horizon, which leads to larger spatial perturbations and, therefore, enhanced amplitudes. However, a small $k_{\mathrm{cr}}/aH$ also reduces the efficiency of ALP production, since the influence of Hubble friction in the equations of motion becomes stronger. Numerically, we find that for $f_\phi \gtrsim 5 \times 10^{15}\,\mathrm{GeV}$, provided that $S_i \sim 10^{18}\,\mathrm{GeV}$, the exponential production of excited ALP quanta becomes ineffective. We take this as a hint that the treatment in the previous section, where we neglected the more stringent relation for efficient growth as well as growth times, over simplified the picture.

Table 2: The parameters used in order to calculate the spectra shown in Fig. 12.

| Benchmark | $m_{S,0}$ [GeV] | $f_\phi$ [GeV] | $m_\phi$ [GeV] | $S_i$ [GeV] |
|---|---|---|---|---|
| 1 | $10^{-1}$ | $1 \times 10^{15}$ | $5 \times 10^{-3}$ | $2 \times 10^{18}$ |
| 2 | $10^{-4}$ | $5 \times 10^{14}$ | $2 \times 10^{-6}$ | $2 \times 10^{18}$ |
| 3 | $10^{-9}$ | $5 \times 10^{14}$ | $3 \times 10^{-11}$ | $2 \times 10^{18}$ |
| 4 | $10^{-12}$ | $5 \times 10^{14}$ | $3 \times 10^{-14}$ | $2 \times 10^{18}$ |
| 5 | $10^{-20}$ | $1 \times 10^{15}$ | $5 \times 10^{-22}$ | $2 \times 10^{18}$ |

### A.4 Calculation of the ALP Fragmentation GW Spectrum

In this section, we provide the calculation of the GW spectrum that is expected from ALP fragmentation. The subsequent discussion is taken from Ref. [99].

At subhorizon scales, the GW power spectrum of a SGWB reads

$$\frac{d\Omega_{\text{GW}}}{d\log k} = \frac{1}{\rho_{\text{tot}}} \frac{M_{\text{Pl}}^2 k^3}{8\pi^2 a^2} \mathcal{P}_{h'}(\mathbf{k}, \tau), \tag{81}$$

with $\langle h'(\mathbf{k}, \tau) h'^*(\mathbf{k}', \tau) \rangle = (2\pi)^3 \mathcal{P}_{h'}(\mathbf{k}, \tau) \delta(\mathbf{k} - \mathbf{k}')$. We may express Eq. (81) as a function of the source, yielding [1]

$$\frac{d\Omega_{\text{GW}}}{d\log k} = \frac{1}{\rho_{\text{tot}}} \frac{k^3}{4\pi^2 a^4 M_{\text{Pl}}^2} \int_{\tau_i}^{\tau} d\tau' d\tau'' a(\tau') a(\tau'') \cos\big(k(\tau' - \tau'')\big) \Pi^2(\mathbf{k}, \tau', \tau''), \tag{82}$$

where $\Pi^2$ is the Unequal Time Correlator (UTC). This object is defined as

$$\langle \Pi_{ij}(\mathbf{k}, \tau) \Pi_{ij}^*(\mathbf{k}', \tau') \rangle = (2\pi)^3 \Pi^2(\mathbf{k}, \tau', \tau'') \delta(\mathbf{k} - \mathbf{k}'), \tag{83}$$

with $\Pi_{ij}$ denoting the anisotropic stress energy. This expression can be evaluated by applying the transverse traceless projectors $\Lambda_{ij}^{kl} = \Lambda_i^k \Lambda_j^l - \frac{1}{2} \Lambda_{ij} \Lambda^{kl}$, with $\Lambda_{ij} = \delta_{ij} - \partial_i \partial_j / \nabla^2$ to the stress energy tensor of the excited ALP modes. We find

$$\Pi_{ij}(\mathbf{k}) = \frac{1}{a^2} T_{ij}^{\text{TT}} = \frac{\Lambda_{ij}^{kl}(\mathbf{k})}{a^2} \int \frac{d^3 q}{(2\pi)^3} q_k q_l \, \phi(\mathbf{q}) \phi(\mathbf{k} - \mathbf{q}). \tag{84}$$

Plugging in the field operators defined by Eq. (66) yields

$$\Pi^2(\mathbf{k}, \tau', \tau'') = 2 \frac{\Lambda_{ij}^{kl}(\mathbf{k})}{a(\tau')^2} \frac{\Lambda_{ij}^{rs}(\mathbf{k})}{a(\tau'')^2} \int \frac{d^3 q}{(2\pi)^3} q_k q_l q_r q_s \, u_q(\tau') u_q(\tau'') u_{|\mathbf{k}-\mathbf{q}|}(\tau') u_{|\mathbf{k}-\mathbf{q}|}(\tau''), \tag{85}$$

where $\langle 0 | \hat{a}(\mathbf{q}) \hat{a}(\mathbf{k} - \mathbf{q}) \hat{a}^\dagger(\mathbf{q}') \hat{a}^\dagger(\mathbf{k}' - \mathbf{q}') | 0 \rangle = 2(2\pi)^6 \delta(\mathbf{k} - \mathbf{k}') \delta(\mathbf{q} - \mathbf{q}')$ has been used in the second step. This expression is further simplified by

$$\Lambda_{ij}^{kl} \Lambda_{ij}^{rs} q_k q_l q_r q_s = \frac{q^4}{2} \sin^4 \theta, \tag{86}$$

where $\theta$ denotes the angle between the two momenta $\mathbf{k}$ and $\mathbf{q}$. The expression for the energy density of the GW per logarithmic frequency then reads

$$\begin{aligned}
\frac{d\Omega_{\text{GW}}}{d\log(k)} &= \frac{1}{\rho_{\text{tot}}} \frac{k^3}{4\pi^2 a^4} \frac{1}{M_{\text{pl}}^2} \int d\tau' d\tau'' a(\tau') a(\tau'') \cos[k(\tau' - \tau'')] \Pi^2(\mathbf{k}, \tau', \tau'') \\
&= \frac{1}{\rho_{\text{tot}}} \frac{k^3}{4\pi^2 a^4} \frac{1}{M_{\text{pl}}^2} \int \frac{d^3 q}{(2\pi)^3} q^4 \sin^4 \theta \left[ |I_c(\mathbf{k}, \mathbf{q}, \tau)|^2 + |I_s(\mathbf{k}, \mathbf{q}, \tau)|^2 \right],
\end{aligned} \tag{87}$$

with

$$I_{c/s}(\mathbf{k}, \mathbf{q}, \tau) = \int_{\tau_i}^{\tau} \frac{d\eta}{a(\eta)} \left\{ \begin{array}{c} \cos(k\eta) \\ \sin(k\eta) \end{array} \right\} u_q(\eta) u_{|\mathbf{k}-\mathbf{q}|}(\eta). \tag{88}$$

Here, the identity $\cos[k(x - y)] = \cos(kx)\cos(ky) + \sin(kx)\sin(ky)$ is used to split the integrals. The integration over the angle $\phi$ is evaluated, while the integration over $\theta$ is replaced

by an integration over $l = |\mathbf{k} - \mathbf{q}| = \sqrt{k^2 - q^2 - 2kq \cos\theta}$. With this, the GW spectrum from excited ALP modes can be written in its final form

$$
\frac{d\Omega_{\text{GW}}}{d\log(k)} = \frac{1}{\rho_{\text{tot}}} \frac{k^2}{16\pi^4 a^4} \frac{1}{M_{\text{pl}}^2} \int_0^\infty dq\, q^5\, \sin^4\theta \int_{|k-q|}^{|k+q|} dl\, l \left[ |I_c(\mathbf{k}, \mathbf{q}, \tau)|^2 + |I_s(\mathbf{k}, \mathbf{q}, \tau)|^2 \right]. \tag{89}
$$

For the numerical calculation of the GW spectrum, the integrals over the momenta are discretized.

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
