# Peer review of "Audible Axions with a Booster: Stochastic Gravitational Waves from Rotating ALPs"

_SciPost Physics, doi:SciPost Phys. 12, 171 (2022)_

## Round 2 · Referee Report · Anonymous · 2022-2-20

Strengths
1-Key ideas explained very clearly
2-Analytical estimates demonstrating the key parameter dependencies are backed up by numerical simulations
3-Testable model addressing open questions in cosmology
Report
The present manuscript studies the generation of gravitational waves, dark matter and a matter-antimatter asymmetry from an axion(-like) particle coupled to dark photons. Extending earlier work on `audible axions' the authors allow for a large initial kinetic energy of the axion, sourced by PQ violating terms in the potential. Large parts of the presented model are testable via future gravitational wave observations, with complementary constraints arising from axion searches. The paper is well written and the key ideas are explained clearly. Analytical estimates demonstrating the key parameter dependencies are backed up by numerical simulations, resulting in a convincing picture. I only have a few questions (see below) before recommending this paper for publication.
Requested changes
1- Eq. (3.13) is UV divergent. How is this quantity regulated to ensure that the UV divergent vacuum contribution does not impact the axion dynamics?
2- If I understand correctly, the radial part of PQ field is taken to decrease due to Hubble friction before becoming constant once reaching the minimum of the Mexican hat potential. With the total PQ charge scaling as $S^2 \dot\theta \sim a^{-3}$, the decrease in $S$ becomes non-trivial once $\dot\theta$ is no longer constant due to energy transfer between the axion motion and the dark photons. Could this significantly impact any of the predictions?
3- With the arguments presented, I'm not entirely convinced that Eq. (5.3) is a lower bound. It rather seems to me like an estimate based on observations in the numerical simulations? It would be nice to have some analytical understanding as to why energy density in the axion field is well described by a kination-like scaling starting at $a_\text{GW}$.
Author: Wolfram Ratzinger on 2022-04-15 [id 2391]
(in reply to Report 1 on 2022-02-20)
We thank the referee for the very positive report. In regards to the questions raised, we have the following replies:
1) Eq.(3.13) is indeed UV divergent. In our simulations it is regulated by a UV cutoff, since we only include modes with big enough momenta to properly cover the band experiencing tachyonic growth. We want to point out though that the modes in the band grow by many orders of magnitude such that they dominate in Eq.(3.13) as long as one does not include modes with momenta many orders of magnitude larger than the tachyonic band. The exact choice of the UV cutoff therefore has a negligible influence on the result if one does not choose it much larger than the momenta in the tachyonic band. We added a corresponding footnote to the manuscript.
2) As explained below Eq.(3.13) the non-trivial radial motion that sets in once the axion slows down brings many uncertainties, since our semi-classical treatment can not be trusted anymore. The features of the GW spectrum that we mainly focus on are however already set when the field first slows down such that we expect them to be reliable. The axion abundance can however only be estimated as we attempted to do in the following sections of the paper.
3) This is a valid remark, indeed our estimate is based on observations of the numerical simulations, and we changed the wording in the text accordingly. In principle we agree that an analytic estimate would be interesting, however a better understanding of the ALP relic abundance would anyways require including the dynamics of the radial mode and the inhomogeneities in the ALP induced by backscattering.
Anonymous on 2022-05-10 [id 2453]
(in reply to Wolfram Ratzinger on 2022-04-15 [id 2391])The authors have taken into account all notes raised in the first review. The paper can now be accepted for publication in SciPost
Author: Wolfram Ratzinger on 2022-04-15 [id 2390]
(in reply to Report 2 on 2022-03-09)We thank the referee for the very positive report. In regards to the questions raised, we have the following replies:
1) As explained in [2] one can think of the non-linear interaction sourcing the GWs as photons colliding to form gravitons. If two photons of the same helicity collide head-on their spins cancel, such that the gravitons spin is only resulting from orbital angular momentum, such that the resulting GWs are unpolarized. This explains why the GW spectrum is only partially polarized even though the photon spectrum is fully polarized. This effect becomes stronger towards the tail of the spectrum, since it is primarily formed from the head-on collision of photons with momenta close to the peak. We added a comment referring the reader to [2].
2) We only include constraints on the ALP-photon coupling, see e.g. the PDG, Fig. 91.1 in the axion chapter. The SN1987A constraints appear to be less robust (according to the PDG) and are therefore not included in our analysis, but would agree with those shown in Refs. [47] and [48].
3) Thanks for pointing out this typo. We corrected it.
4) We have added a discussion of this point at the end of section 2.5.

---

## Round 2 · Referee Report · Anonymous · 2022-3-9

Strengths
1- Interesting framework to produce gravitational waves detectable by the current and future experiments, which could also address the dark matter and baryogenesis problem.
Weaknesses
1- Oversimplification of the evolution of the complex field $P$ (lack of an UV-complete model).
Report
The present paper discusses the production of gravitational waves generated by dark photons tachyonic instability, due to the evolution of axions or axion-like particles (ALP). Here the authors use the kinetic misalignment mechanism instead of the conventional misalignment mechanism. They present parameter space where tachyonic production of dark photons becomes efficient and the GW spectra for the different benchmark parameter points. Furthermore, the authors estimate relic abundance of the ALP and when the ALP is the QCD axion, with the possibility of produce the baryon asymmetry by the axiogenesis mechanism. Finally, the authors find it is possible that the dark photon can be the DM too.
The paper builds upon earlier work by the authors, it is well written and summary of the relevant mechanisms at work, and leads to interesting predictions that may serve as a benchmark in upcoming gravitational wave searches. It is therefore suitable for publication in SciPost. However, before it proceeds to publication, I would like to ask the authors to perform a minor revision of their manuscript in which they address the following questions and comments.
Requested changes
1-In Figure 4 left, the authors show the GW spectrum of the plus and minus helicities. They explain that one helicity is enhance over the other depending on the sign of the axion velocity. Naively, I would have expected the difference to be bigger, meaning almost zero for the minus helicity, not ~$\mathcal{O}(100)$. It would be nice to understand where this difference might be coming from analytically.
2-In figure 7 the authors show the excluded region for $f_{\phi} \gtrsim 3.4 \times 10^{7} \textrm{GeV} $ due to stellar constrains, but looking into references [47] and [48] these constraints are more stringer. I wonder why there are different.
3- In Table 1 the saxion mass should be $m_{S,0}$ instead of $m_{s}$
4-For very light saxion masses, the coupling $\lambda$ would be subject to quantum correction which will imposes some constrain in $y_{\psi}$. I think is worth of explaining this point.

---

## Round 3 · List of Changes

Minor changes as requested. See correspondence with referees for details.

---

## Editorial Decision

published